# Reversible gating of singlet fission by tuning the role of a charge-transfer state

Yifan Bo [1], Yuxuan Hou [2,4], Dominic A. X. Lavergne[2], Timothy Clark [3], Michael J. Ferguson [2], Rik R. Tykwinski [2] ✉ & Dirk M. Guldi [1] ✉

Stimulus-responsive triplet excited states and multiexcitonic logic gates have garnered increasing interest. Singlet fission is an efficient multiple exciton generation process, in which one singlet converts into two triplets. Singlet fission is, however, rarely reported to be switchable by external stimuli. Here we design a *meta*-diethynylphenylene-linked tetracene dimer featuring pyridyl endgroups that function as an acid/base-responsive switch, enabling the reversible modulation of singlet fission. In its neutral form, the inter-chromophore charge-transfer state facilitates singlet fission and promotes the formation of a correlated triplet-pair state. Upon treatment with acid, protonation of the pyridyl nitrogens generates a more strongly electron-accepting pyridinium, leading to an intra-chromophore charge-transfer state, which inhibits singlet fission. Finally, an IMPLICATION logic gate is constructed by using acid and base as inputs and monitoring the formation of triplet excited states based on singlet fission.

The excited dynamics of molecules are sensitive to environmental stimuli such as pH, temperature, viscosity, etc., resulting in changes in structural, electronic, or spectroscopic features[1-4]. Such a response can be used to construct a molecular switch and, further, complete molecular logic gates (MLGs) as they convert physical or chemical inputs into spectroscopic outputs. The development of MLGs has received extensive attention, starting with the ground-breaking research on fluorescent chemosensors in 1993[1,5-12]. Most investigations of MLG focus on changes in the photoluminescence, which typically relies on the radiative decay of a singlet excited state as output following an external stimulus as input. The benefit lies in the reversible switching of singlet excited-state fluorescence between two distinct mechanisms: energy transfer versus electron transfer[13-15]. Triplet excited-state switching[16] and multi-excitonic logic devices[17,18] remain largely unexplored.

Singlet fission (SF) can, in principle, convert a singlet exciton into two triplet excitons. Thus, the maximum triplet quantum yield for SF is 200%[19-23]. SF is a well-known multiple-exciton generation process that is of interest not only for efficient solar energy harvesting[24-26] but also

for photocatalysis[27], photodynamic oncotherapy[28], large optical nonlinearity[29-32]. Observing a high-spin quintet state in SF, which is undoubtfully a rarity in organic materials, recently invokes the potential for SF to be applied in quantum information science[33-35]. In short, designing a molecular logic gate based on an SF switch would not only enable on/off control over multi-excitonic processes but also open up new avenues for exploring triplet excited-state switching. Therefore, it would be beneficial to realize a SF switch that responses to environmental stimuli and that transforms SF beyond basis research to, for example, molecular computers, molecular mechanisms, and biology as achieved by traditional molecular switches[1,3,4,36].

Efficient SF requires two or more chromophores with both suitable energetics and sufficient electronic coupling[19-23]. For the former, the energy of the singlet exciton must be greater than or nearly equal to twice the triplet exciton energy. Acenes, especially pentacene[37-41] and tetracene[42-45], are excellent candidates for either exothermic or slightly endothermic SF, respectively. For the latter criterion, inter-chromophore interactions (coupling) should be present. In acene oligomers, especially dimers, in which acenes are covalently linked to

[1]Department of Chemistry and Pharmacy & Interdisciplinary Center for Molecular Materials (ICMM), FAU Profile Center Solar, Friedrich-Alexander-Universität Erlangen-Nürnberg, Erlangen, Germany. [2]Department of Chemistry, University of Alberta, Edmonton, AB, Canada. [3]Department of Chemistry and Pharmacy & Computer-Chemie-Center (CCC), Friedrich-Alexander-Universität Erlangen-Nürnberg, Erlangen, Germany. [4]Present address: Department of Chemistry, University of Copenhagen, Copenhagen, Denmark. ✉e-mail: rik.tykwinski@ualberta.ca; dirk.guldi@fau.de

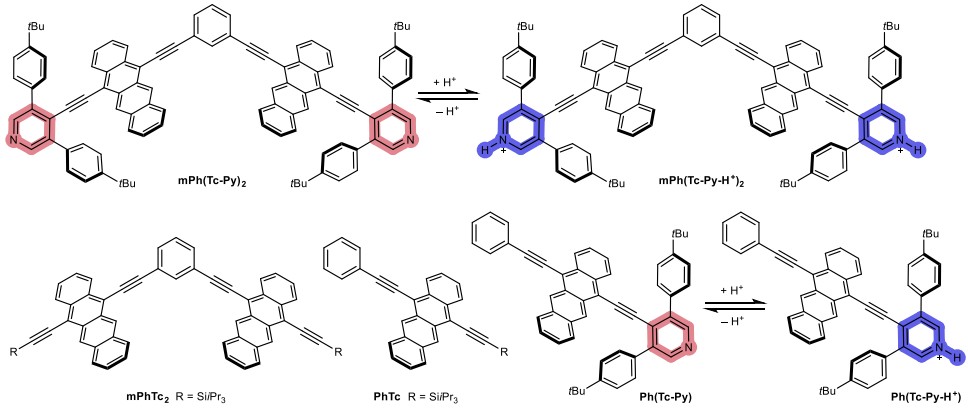

**Fig. 1 | The design of our work by merging tetracene dimer with pyridyl endgroups.** Chemical equilibrium of mPh(Tc-Py)$_2$/mPh(Tc-Py-H$^+$)$_2$ and Ph(Tc-Py)/Ph(Tc-Py-H$^+$); molecular structures of mPhTc$_2$ and PhTc.

each other via a spacer, inter-chromophore electronic coupling is tailorable and tunable[39].

Two mechanisms for SF have been widely discussed in the literature. First, a direct mechanism is possible, in which the initial (S$_1$S$_0$) state is directly converted into a multi-exciton $^1$(T$_1$T$_1$) state via a concerted two-electron process[19,46,47]. Secondly, an indirect mechanism is also possible, in which a charge-transfer (CT) state plays a decisive role[40,43,48–55]. In the indirect mechanism, CT states have been demonstrated to act as virtual or real intermediates to affect SF[40,49,52], and the relative energies of (S$_1$S$_0$), $^1$(T$_1$T$_1$), and CT state are important. Efficient SF has also been suggested to proceed through the electronic superposition of several states including (S$_1$S$_0$), $^1$(T$_1$T$_1$), and CT states[43,48,50,51,53,54]. However, a CT state might serve as a trap state that inhibits SF in some cases[48,49,56]. For example, in polar solvents, CT states are subject to energetic stabilization. In the presence of a trap, either the (S$_0$S$_0$) state or the charge-separated state is formed rather than promoting the formation of $^1$(T$_1$T$_1$). Adjusting the energy of the CT state in relation to that of (S$_1$S$_0$) and $^1$(T$_1$T$_1$) states can thus either promote or inhibit SF.

Previous research efforts have been dedicated to controlling SF, and, in turn, these studies form the foundation for design of a SF switch. In the solid state, SF is mostly investigated in monomers with variable molecular packing through molecular engineering[57–62] or conditioned fabrication[63,64]. In the context of molecular engineering, tunable SF is investigated in different stacking geometries. These are adjusted by functionaled SF chromophores, such as alkyl-substituted terrylenes[59] and diketopyrrolopyrroles[57,60]. For conditioned fabrication, SF processes are compared in amorphous films, polycrystalline films, and single crystals. Moreover, changes in SF in films are reported upon varying the aggregation through varying the solvent or different ratios of the doped polymer matrix[62]. Controlling the packing modes and achieving homogeneity in the solid state is, per se, challenging and limits utility as a platform for the design of SF switches. In contrast, molecular dimers offer advantages such as studies in solution with precise control and tuning of the molecular structure, as well as the distance, geometric relationship, and electronic coupling between two chromophores. For example, Alvertis and coauthors demonstrated that the transition between an incoherent and a coherent mechanism for SF in an orthogonal tetracene dimer is modulable by varying the solvent environment[65]. Wasielewski and coworkers work with terrylenediimide dimers noted that the CT state in polar solvents not only acted as a trap state but also suppressed SF[49]. Guldi and coworkers developed the concept of SF switching by employing a diamantane spacer placed between two pentacenes. They showed that SF could be toggled 'on' or 'off' by altering the substitution pattern[37]. Thus, they realized a SF switch through two distinct dimeric isomers. Altogether, these studies demonstrate that SF is sensitive to both molecular structure and the environment. To the best of our knowledge, however, a reversible SF switch in a molecular system, which can be further applied to molecular logic gates, has never been reported.

We have now opted for a dimer that enables SF via an indirect mechanism. Furthermore, the dimer bears a chemical probe in the molecular framework designed to operate as IMPLICATION gates using triplet excited states from SF as the output. Based on our previous work[43], triplet quantum yields of over 100% in the cross-conjugated *meta*-diethynylphenylene tetracene dimer mPhTc$_2$ are mediated by a mixed state with contributions from (S$_1$S$_0$), (T$_1$T$_1$), and CT states (Fig. 1). The pyridine moiety has often served as an electron acceptor in the design of push-pull systems due to its well-known electron-accepting nature[66–68]. Moreover, the nitrogen atom is a Brønsted and Lewis base, and, for example, can be easily protonated. Notably, upon protonation, the resulting pyridinium salt is a stronger electron acceptor due to its positive charge, while the aromaticity is retained.

In this contribution, a tetracene dimer mPh(Tc-Py)$_2$ with pyridyl endgroups has been designed and synthesized (Fig. 1). The tetracene dimer mPh(Tc-Py)$_2$ is SF-active, and pyridine is the acid/base responsive probe. Protonation of the probe results in the intra-chromophore charge transfer between the electron-donating tetracene and electron-accepting pyridinium that, in turn, inhibits SF. Upon sequential exposure of mPh(Tc-Py)$_2$ to trifluoroacetic acid and triethylamine (TFA and TEA, respectively), we note a reversible 'off-on' response. Thus, the system is used to exhibit conceptual operation as an IMPLICATION gate with acid and base as inputs and SF-born (T$_1$T$_1$) as output. Finally, the corresponding monomer Ph(Tc-Py) has been synthesized and investigated in comparison to mPh(Tc-Py)$_2$ to better describe SF/triplet excited-state switching.

## Results
### Synthesis
The *meta*-diethynylphenylene-linked tetracene dimer mPh(Tc-Py)$_2$ and its corresponding monomer Ph(Tc-Py) were synthesized by the desymmetrization of 5,12-naphthacenequinone via stepwise nucleophilic acetylide addition (Supporting Information). Briefly, the lithiation of the corresponding pyridylacetylene provided the acetylide that was added to a solution of 5,12-naphthacenequinone in dry THF at −78 °C. In the same pot, lithiated 1,3-diethynylbenzene or ethynylbenzene was added to complete the carbon skeleton. Reductive aromatization of the intermediate diol with SnCl$_2$ • 2H$_2$O, followed by purification by column chromatography with silica gel and recrystallization, gave the desired products mPh(Tc-Py)$_2$ and Ph(Tc-Py) in 59% and 76% yield, respectively. Synthetic and characterization details can be found in the Supporting information.

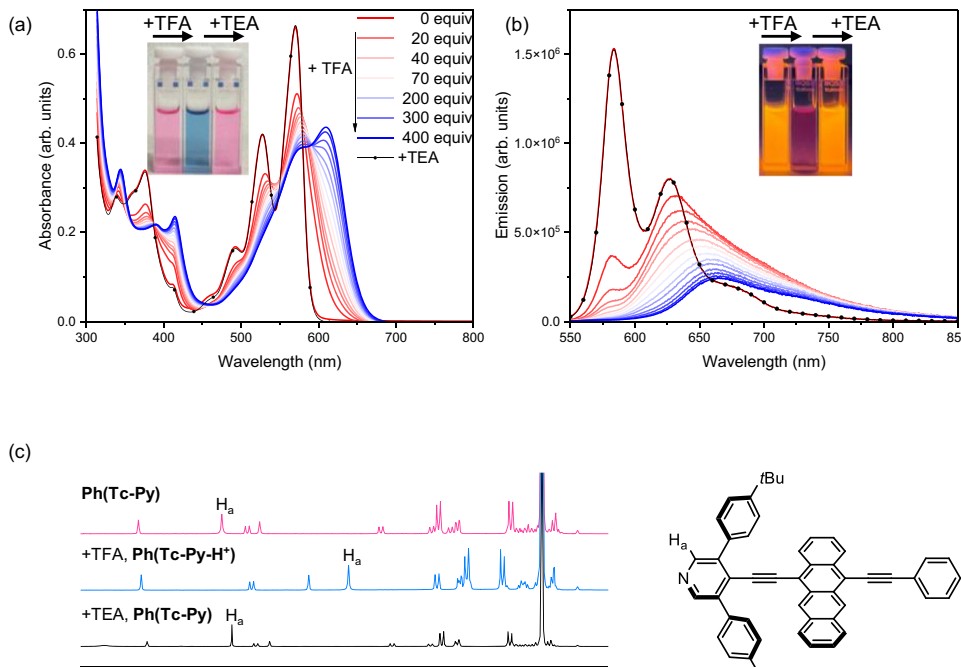

**Fig. 2 | The switchable behavior on the steady-state photophysical properties and proton nuclear magnetic resonance spectroscopy ($^1$H NMR). a** Changes of absorption and (**b**) the corresponding fluorescence of mPh(Tc-Py)$_2$ in toluene ($1 \times 10^{-5}$ m) upon the addition of TFA (from 0 to ca. 400 equivalents); dotted black line is after subsequent addition of TEA (ca. 420 equivalents). **c** Comparison of the aromatic region of $^1$H NMR spectra of Ph(Tc-Py) upon adding an excess of TFA and an excess of TEA in C$_6$D$_6$; proton signals of pyridyl/pyridinium group are labeled as H$_a$. Source data are provided as a Source Data file.

## Steady-state characterization

Steady-state absorption spectra in toluene are shown in Supplementary Fig. 5 and summarized in Supplementary Table 1. The monomer Ph(Tc-Py) displays well-defined absorption maxima at 489, 523, and 564 nm, which are red-shifted by ca. 20 nm due to the π-extension when compared to those observed for PhTc with tri*iso*propylsilyl (*i*Pr$_3$Si) endgroups[43]. The absorption spectra of mPh(Tc-Py)$_2$ reveal even greater red-shifted absorptions at 491, 528, and 570 nm, respectively, and notably the molar extinction coefficient (ε) associated with the tetracene 0−0* transition of mPh(Tc-Py)$_2$ is ε = 61,700 M$^{-1}$cm$^{-1}$, which is less than twice that of Ph(Tc-Py) with ε = 33,700 M$^{-1}$cm$^{-1}$. The absorption data substantiate tetracene-tetracene electronic interactions in dimer mPh(Tc-Py)$_2$, as previously documented for mPhTc$_2$[43]. Notably, the absorption spectrum of mPh(Tc-Py)$_2$ is well-defined without any discernable broadening compared to that of Ph(Tc-Py), indicating the rigid and homogeneous conformation of mPh(Tc-Py)$_2$ in solution.

The fluorescence spectra of Ph(Tc-Py) and mPh(Tc-Py)$_2$ show red-shifted emission maxima, with maxima at 578 and 583 nm, respectively, compared to PhTc and mPhTc$_2$, with maxima at 550 and 555 nm, respectively. Compared with the high fluorescence quantum yields (Φ$_F$ = 80%) that are measured under ambient conditions for Ph(Tc-Py), mPh(Tc-Py)$_2$ displays a much lower value of Φ$_F$ = 28%. The lower fluorescence quantum yield for mPh(Tc-Py)$_2$ is fully consistent with a competing non-radiative deactivation pathway, i.e., SF, which dominates excited-state deactivation (Supplementary Table 1). In the more polar solvent benzonitrile, absorption and fluorescence of Ph(Tc-Py) and mPh(Tc-Py)$_2$ are subtly red-shifted without, however, any visible changes in terms of vibronic fine-structure (Supplementary Fig. 6 and Supplementary Table 1).

## Acidochromic properties

The optimized structures and the molecular electrostatic potential (MEP) maps of Ph(Tc-Py) and mPh(Tc-Py)$_2$, obtained using density functional theory (DFT) at the B3LYP-GD3BJ/def2svp level of theory, are shown in Supplementary Fig. 7. Significant negative electrostatic potentials are localized on the nitrogen atoms with values of −0.06 a.u. corresponding to the attraction of protons by areas of concentrated electron density.

The modulation of steady-state photophysical properties was explored experimentally via protonation with TFA (Fig. 2a, b; Supplementary Fig. 8, and Supplementary Table 1). Intriguingly, the color of Ph(Tc-Py) solutions changed from reddish to blueish once TFA was added (Supplementary Fig. 8). Throughout the TFA titration, an absorption at 611 nm increased in intensity at the expense of that at 564 nm, indicating that protonated Ph(Tc-Py-H$^+$) features a smaller band gap than its un-protonated form, Ph(Tc-Py). We note that along with the absorption changes, the fluorescence red-shifts to 673 nm and its quantum yield reaches 20%. Notably, throughout all the TFA titrations the spectra lose the vibrational fine structure. The acidochromic behavior of mPh(Tc-Py)$_2$ is quite similar (Fig. 2a, b). In the protonated form, the absorption and fluorescence of mPh(Tc-Py-H$^+$)$_2$ are red-shifted to 615 and 671 nm, respectively. It is, however, impossible to differentiate between mono-protonated mPh(Tc-Py)(Tc-Py-H$^+$) and fully protonated mPh(Tc-Py-H$^+$)$_2$ from titration measurements using absorption spectroscopy. Thus, for any further considerations we refer to mPh(Tc-Py-H$^+$)$_2$, which is formed after the addition of an excess TFA. Proton responsive Ph(Tc-Py) and mPh(Tc-Py)$_2$ have been checked in reversible titrations with TFA and followed by triethylamine (TEA) (dotted black line in Supplementary Fig. 8 and Fig. 2). Full protonation of Ph(Tc-Py) and mPh(Tc-Py)$_2$ is also verified by means of noticeable shifts for the pyridyl proton signals in the $^1$H NMR spectra (Fig. 2c and Supplementary Fig. 9). The pyridyl protons (H$_a$) for Ph(Tc-Py) are identified with ease in the aromatic region as a singlet with an integration of 2H. Both absorption and fluorescence of Ph(Tc-Py) and mPh(Tc-Py)$_2$ are quantitatively reinstated upon adding TEA to Ph(Tc-Py-H$^+$) and mPh(Tc-Py-H$^+$)$_2$, respectively. The reversibility is not limited to a single cycle of TFA and TEA addition, but is consistently realized in several cycles of alternate additions (Supplementary Fig. 10). Moreover, cycles of protonation and deprotonation are also visible to the naked eye (Fig. 2a, b). It should be noted that the accumulation of

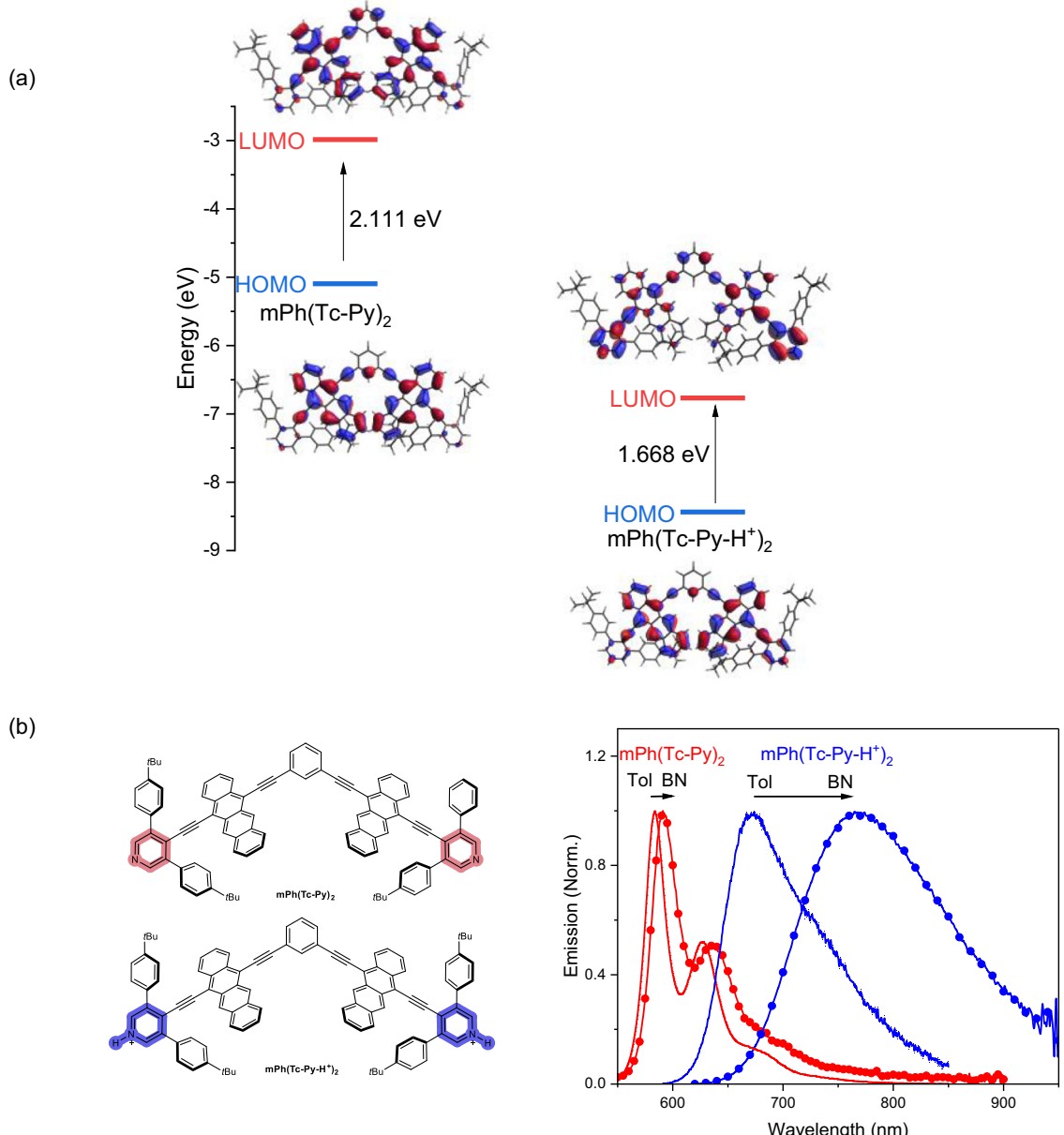

**Fig. 3 | Frontier molecular orbitals and solvent-dependent absorption and emission of tetracene dimer before and after protonation. a** Graphic representation of frontier molecular orbitals and respective energies of mPh(Tc-Py)$_2$ and its protonated congeners mPh(Tc-Py-H$^+$)$_2$ as determined by DFT at the B3LYP-GD3BJ/def2-SVP level. **b** The fluorescence of mPh(Tc-Py)$_2$ and mPh(Tc-Py-H$^+$)$_2$ in toluene (solid red and blue lines, respectively) and benzonitrile (dotted red and blue lines, respectively). Source data are provided as a Source Data file.

byproducts generated from acid and base additives is likely to hinder the complete resetting of chemical arithmetic systems, thereby disrupting the overall reversibility that is required for cyclic operations[67,69,70].

The electronic structures of the neutral and protonated states were optimized at the B3LYP-GD3BJ/def2svp level of density-functional theory to help appreciate changes in electronic properties upon protonation, (DFT, Fig. 3a and Supplementary Fig. S11). The highest occupied (HOMO) and lowest unoccupied (LUMO) molecular orbitals of the neutral forms are both delocalized across Ph(Tc-Py) and mPh(Tc-Py)$_2$. The protonated forms leave the HOMOs relatively unchanged, but the LUMOs become more localized on the pyridinium. This suggests an intramolecular charge transfer after protonation. Notably, protonation reduces the HOMO-LUMO gap from 2.15 eV for Ph(Tc-Py) and 2.11 eV for mPh(Tc-Py)$_2$ to 1.65 eV for Ph(Tc-Py-H$^+$) and 1.67 eV for mPh(Tc-Py-H$^+$)$_2$, which is consistent with the red-shifted

absorption seen upon protonation. A similar trend has been observed in previous investigations of pyridyl PAHs[71,72]. The underlying nature, that is, an intra-chromophore CT (intra-CT) from tetracene to pyridinium, is corroborated by a fluorescence that is sensitive to the solvent polarity (Supplementary Table 1; Fig. 3b and Supplementary Fig. 12). Compared to the subtle red-shifts seen for the fluorescence of Ph(Tc-Py) and mPh(Tc-Py)$_2$ in benzonitrile versus toluene, distinct red-shifts of nearly 100 nm evolve for Ph(Tc-Py-H$^+$) and mPh(Tc-Py-H$^+$)$_2$. The Stokes shifts for the protonated species in benzonitrile that go hand-in-hand with red-shifted fluorescence maxima are significant. Moreover, as seen in excitation-emission maps that were recorded for mPh(Tc-Py-H$^+$)$_2$ in toluene (Supplementary Fig. 13), intra-CT emission dominates. This occurs regardless of the photoexcitation wavelength and locally-excited emission is not observed even under high-energy photoexcitation. This emission behavior indicates that the formation of a low-lying intra-CT state is highly competitive.

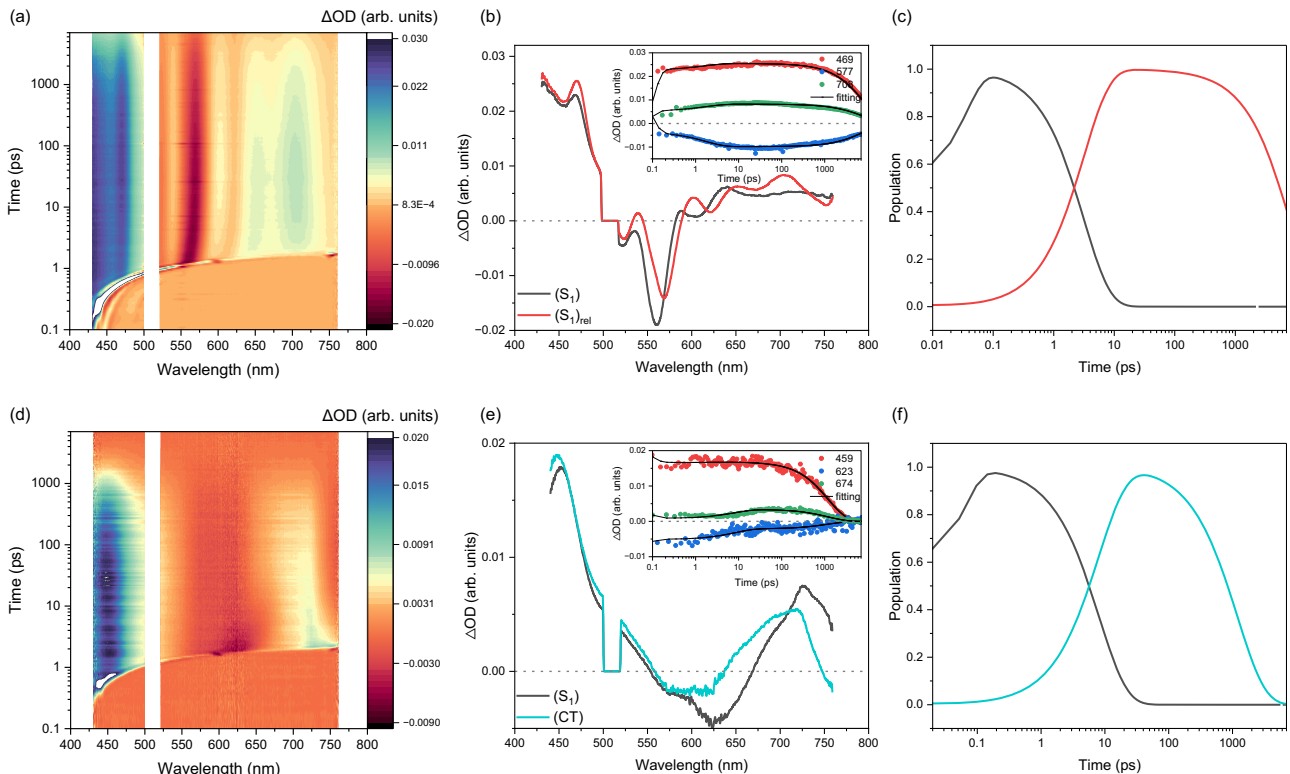

**Fig. 4 | Femtosecond transient absorption of Ph(Tc-Py) and Ph(Tc-Py-H⁺) in toluene.** Femtosecond transient absorption spectroscopy of (**a**–**c**) Ph(Tc-Py) and (**d**–**f**) Ph(Tc-Py-H⁺) in argon-saturated toluene at room temperature. **a**, **d** Heat map of the recorded differential absorption spectra at various time delays between 0 to 7000 ps after photoexcitation at 510 nm. **b**, **e** Evolution-associated spectra (EAS) and insert that depicts time absorption profiles as well as corresponding fits of selected wavelengths (see Figure legend for details). **c**, **f** The relative population of the respective species with colors correlating with the evolution-associated spectra (EAS). Source data are provided as a Source Data file.

## Excited-state dynamics

The acidochromic effects on the excited-state dynamics of Ph(Tc-Py) are established by performing femtosecond and nanosecond transient absorption spectroscopies (fs-TAS and ns-TAS) in toluene and benzonitrile before and after adding TFA. The raw data and global analyses are collected in Fig. 4a–c and Supplementary Figs. 14–19. In the case of Ph(Tc-Py) in toluene (Fig. 4a–c; Supplementary Figs. 14, 15 and Supplementary Table 3), photoexcitation at 510 nm was used to generate the first singlet excited ($S_1$) state. This ($S_1$) is characterized by ground-state bleaching (GSB) in the range of 500–580 nm with minima at 524 and 560 nm, and by excited-state absorption (ESA) in the range of 400–500 nm with maxima at 420 and 470 nm, and in the range of 580–800 nm and 830–1300 nm with maxima at 880, 1010, and 1160 nm (Fig. 4a and Supplementary Fig. 14a). Moreover, the ESA at 580–800 nm overlaps with stimulated emission (SE) and displays a minimum at ca. 610 nm. The fs-TAS and ns-TAS data of Ph(Tc-Py) in toluene require the use of a kinetic model with two sequential species (Fig. 4b, Supplementary Figs. 14b and 15b). The kinetic model for the fs-TAS data involves ($S_1$), which undergoes solvent reorganization to afford ($S_1$)$_{rel}$ within 3.1 ps and then completely reinstates the ground state ($S_0$) within 8.2 ns. A lifetime of 8.2 ns matches well the corresponding TCSPC lifetimes in toluene. Changing to the more polar solvent benzonitrile, TAS shows only minor changes compared to the results in toluene (Supplementary Figs. 16 and 17, and Supplementary Table 2). For example, slightly red-shifted GSB and SE, together with a larger solvent reorganization, lead to a lifetime of 14.6 ps for ($S_1$) due to the high viscosity of the solvent. It takes an additional 7.9 ns before ($S_1$)$_{rel}$ relaxes back to the ground state in benzonitrile. From the fluorescence quantum yields and lifetimes of ($S_1$)$_{rel}$, the radiative ($k_r$) and nonradiative ($k_{nr}$) rate constants of Ph(Tc-Py) in different solvents are determined (Supplementary Table 2). Similar to unsubstituted

tetracene (Tc) and other tetracene monomers with $k_r$ in the range of $3$–$10 \times 10^7 \, s^{-1}$[73,74], Ph(Tc-Py) exhibits radiative $k_r$ of $9.7 \times 10^7$ and $11.0 \times 10^7 \, s^{-1}$ in toluene and benzonitrile, respectively. Constants $k_{nr}$ of Ph(Tc-Py) are $2.4 \times 10^7$ and $1.4 \times 10^7 \, s^{-1}$ in toluene and benzonitrile, respectively, which are one order of magnitude lower than that of unsubstituted tetracene ($1$–$3 \times 10^8 \, s^{-1}$)[73,74]. Changes in $k_{nr}$ are attributed to the absence of intersystem crossing (ISC) in Ph(Tc-Py), where the yields of ISC of Tc are larger than $0.62$[74]. As a result, Ph(Tc-Py) shows stronger fluorescence than Tc regardless of the solvent. For Ph(Tc-Py), $k_{nr}$ is dominated by internal conversion ($k_{IC}$). Notably, $k_{IC}$ of Ph(Tc-Py) varies as a function of solvents and is higher in toluene than in benzonitrile, but remains around $10^7 \, s^{-1}$, similar to $k_{IC}$ of Tc. Differences in terms of $k_{nr}$ of Ph(Tc-Py) in toluene ($2.4 \times 10^7 \, s^{-1}$) and benzonitrile ($1.4 \times 10^7 \, s^{-1}$) likely stem from solvent effects. In a more viscous benzonitrile, the restriction of intramolecular rotation reduces $k_{IC}$ ($k_{nr}$) and, thereby increases the fluorescence quantum yields. This phenomenon is consistent with observations made for other fluorescent materials[75–77].

The excited-state dynamics of Ph(Tc-Py-H⁺) displays a different picture compared to Ph(Tc-Py), regardless of the solvent (Fig. 4d–f, Supplementary Figs. 18 and 19, and Supplementary Table 3). Notably, the excited-state dynamics of Ph(Tc-Py-H⁺) are fully deconvoluted within fs-TAS in the range of 0–7000 ps and best fit with a sequential model based on two species. After photoexcitation at 510 nm, ($S_1$) is formed instantaneously with GSB in the range from 550 to 670 nm and ESA in the range from 430 to 550 nm and from 670 to 1300 nm with maxima at 450 725, 950, and 1225 nm. In toluene, the second species is formed after 8.2 ps with ESA in the range of 640 to 700 nm and 800 to 1200 nm. In addition, a new negative signal is noted at around 755 nm (Fig. 4e and Supplementary Fig. 18b). The former ESA is similar to the one-electron reduced form of bipyridinium[78], while the latter matches

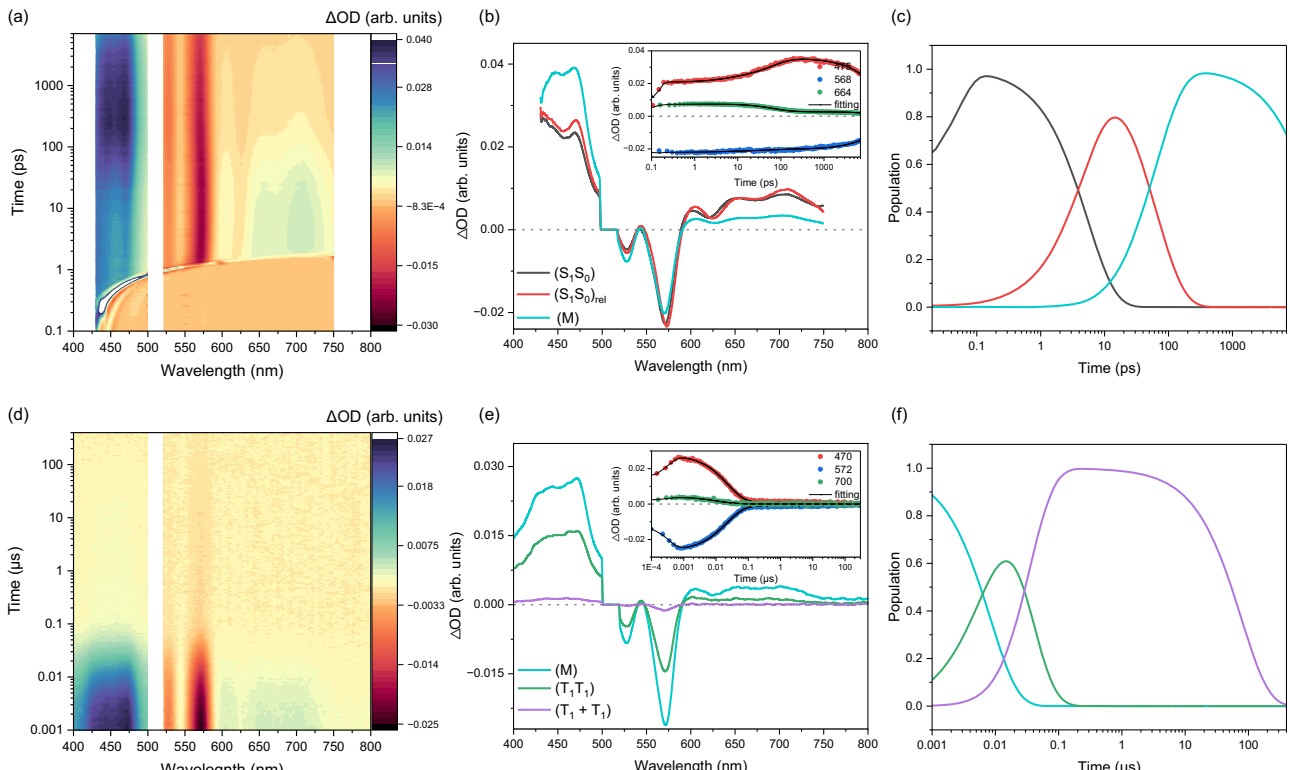

**Fig. 5 | Femtosecond and nanosecond transient absorption of mPh(Tc-Py)₂ in toluene. a–c** Femtosecond and (**d–f**) nanosecond transient absorption spectroscopy of mPh(Tc-Py)₂ in argon-saturated toluene at room temperature. **a** Heat map of the recorded differential absorption spectra at various time delays between 0 to 7000 ps after photoexcitation at 510 nm. **b** Evolution-associated spectra (EAS) and insert that depicts time absorption profiles as well as corresponding fits of selected wavelengths (see Figure legend for details). **c** The relative population of the respective species with colors correlating with the evolution-associated spectra (EAS). **d** Heat map of the recorded differential absorption spectra at various time delays between 0 to 330 μs after photoexcitation at 510 nm. **e** Evolution-associated spectra (EAS) and insert that depicts time absorption profiles as well as corresponding fits of selected wavelengths (see Figure legend for details). **f** The relative population of the respective species with colors correlating with the evolution-associated spectra (EAS). Source data are provided as a Source Data file.

the one-electron oxidized form of tetracene[43,79,80]. As such, we assign this second species as an intra-chromophore CT (intra-CT) state, representing the charge transfer between pyridinium and tetracene (Tc^δ+-Py-H). The negative signal arises from the overlap between SE and ESA and is, per se, emissive. In line with the discussions regarding the steady-state experiments (vide supra), the fluorescent state of Ph(Tc-Py-H⁺) is CT in nature. The second species has a lifetime of 1.2 ns and then recovers later to the ground state, providing another piece of evidence supporting the formation of an intra-CT state. The lifetime of 1.2 ns is analogous to that of the mono-exponential fitting of the fluorescence decay. Changing to the more polar solvent, benzonitrile, an intra-CT state is formed within 10.3 ps from (S₁) and exhibits more pronounced signatures of an intra-CT state, namely the spectral characteristics of the one-electron reduced form of bipyridinium next to the one-electron oxidized form of tetracene (Supplementary Fig. 19). It is observed that the intra-CT state in Ph(Tc-Py-H⁺) forms rapidly within a few picoseconds, regardless of the solvent. This rapid formation is attributed to the strong coupling between the pyridinium and tetracene moieties through the acetylene spacer, which enables an adiabatic charge transfer that follows solvent relaxation dynamics[81–83]. In the more viscous solvent benzonitrile, slower structural relaxation and solvation dynamics lead to a slightly delayed formation of the intra-CT state. Subsequently, in benzonitrile, the intra-CT state is shorter-lived and decays back to the ground state within 445.7 ps. Charge recombination (CR) in Ph(Tc-Py-H⁺) is slow and is subject to a strong polarity dependence. The lifetime of intra-CT state in non-polar toluene (1.2 ns) is thus longer than in more polar benzonitrile (445.7 ps). Overall, CR occurs diabatically. In polar solvents like benzonitrile, the smaller energy gap between the intra-CT state and the

ground state increases the driving force for charge recombination and expedites it. In summary, the stronger electron-accepting character of pyridinium activates the formation of the intra-CT state in Ph(Tc-Py-H⁺), which impacts the excited-state dynamics.

Finally, analyses of the fs- and ns-TAS data for mPh(Tc-Py)₂ and mPh(Tc-Py-H⁺)₂ were carried out to study the acidochromic effects on the multiexciton state produced in SF (Figs. 5 and 6; Supplementary Figs. 21–25 and 28–32 and Supplementary Table 3). Immediately after photoexcitation of mPh(Tc-Py)₂, the characteristics of (S₁), analogous to those for Ph(Tc-Py) in toluene, are observed (Fig. 5a). Subsequently, a new ESA maximum at 470 nm grows, while the ESA at 600 to 750 nm decreases in intensity. Importantly, the new ESA signal resembles the triplet excited state (T₁) that is found in triplet-triplet sensitization measurements[84] (vide infra) and decays on the ns-TAS time range (Fig. 5d and Supplementary Fig. 20). Similar to previous investigations on mPhTc₂[43], a sequential model with five species is required to fit the fs- and ns-TAS data (Fig. 5 and Supplementary Figs. 23a–c and 25a; Supplementary Table 3). The first and second species, which have the characteristics of (S₁) for Ph(Tc-Py), are ascribed to the singlet excited state before (S₁S₀) and after relaxation (S₁S₀)rel. The (S₁S₀)rel state is short-lived, with a lifetime of only 64.5 ps, and it then transforms into a third species. In particular, the third species is not fully deconvoluted by fs-TAS and is identical to the first species in ns-TAS. In addition to the characteristics of (S₁) in the form of an ESA in the 600–750 nm range, additional ESA in the range of 400 to 500 nm with a 470 nm maximum is discernable (Fig. 5b, e; and Supplementary Fig. 21). To identify these newly formed signatures, triplet-triplet sensitization measurements were performed[84], using PdPc(OBu)₈ as photosensitizer[43,85,86] and photoexcitation at 730 nm (Supplementary Fig. 20). Importantly, ESA at

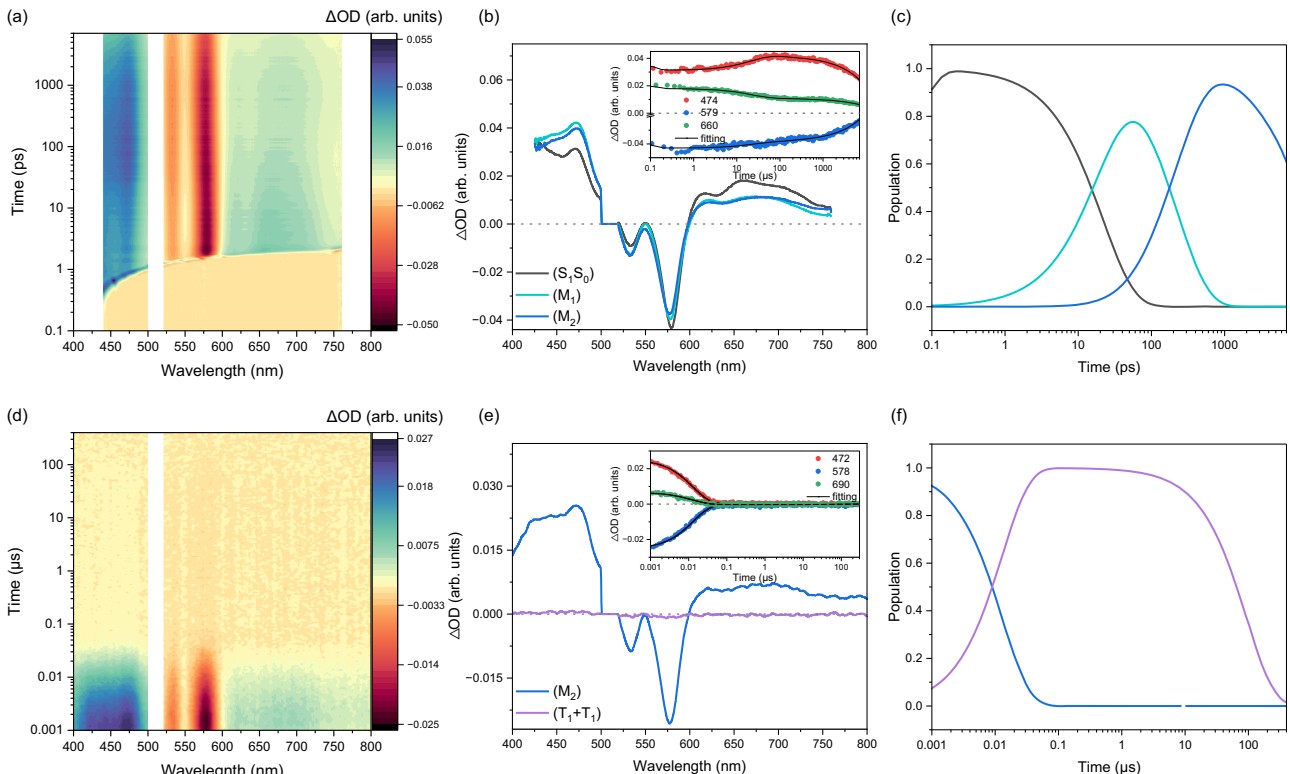

**Fig. 6 | Femtosecond and nanosecond transient absorption of mPh(Tc-Py)₂ in benzonitrile. a–c** Femtosecond and (**d–f**) nanosecond transient absorption spectroscopy of mPh(Tc-Py)₂ in argon-saturated benzonitrile at room temperature. **a** Heat map of the recorded differential absorption spectra at various time delays between 0 and 7000 ps after photoexcitation at 510 nm. **b** Evolution-associated spectra (EAS) and insert that depicts time absorption profiles as well as corresponding fits of selected wavelengths (see Figure legend for details). **c** The relative population of the respective species with colors correlating with the evolution-

associated spectra (EAS). **d** Heat map of the recorded differential absorption spectra at various time delays between 0 and 330 μs after photoexcitation at 510 nm. **e** Evolution-associated spectra (EAS) and insert that depicts time absorption profiles as well as corresponding fits of selected wavelengths (see Figure legend for details). **f** The relative population of the respective species with colors correlating with the evolution-associated spectra (EAS). Source data are provided as a Source Data file.

400 to 500 nm is consistent with that of (T₁) from the triplet-triplet sensitization measurements. Moreover, the spectral similarities between the sensitized (T₁) and the spectroscopic signatures of both the fourth and fifth species in the ns-TAS are striking. In line with previous studies[43,50,53,54,87,88], the third species is a mixed state (M), which is a superposition of (S₁S₀)ᵣₑₗ, CT state, and the triplet pair state (T₁T₁). Here, the CT state is the charge transfer between two tetracenes; we ascribe it to the interchromophore CT (inter-CT) state. Notably, the polarity-dependent behavior of the mixed state suggest that the inter-CT state plays a role in the SF process (vide infra–the analysis of the excited state dynamics in benzonitrile). Furthermore, distinct signatures of inter-CT in the range of 800−1200 nm are clearly observed in the polar solvent benzonitrile. However, in toluene, the contributions of inter-CT in (M) are weak due to the higher energy of inter-CT in apolar solvents[49–51]. Additionally, the spectral features of inter-CT overlap with the singlet and triplet excited-state absorptions, making its unambiguous identification in toluene challenging. Taking this into account, along with the solvatochromic nature of (M) and the presence of both singlet and triplet excited signatures in the EAS of (M), we conclude that the electronic nature of (M) by the mixing of states, consistent with previous investigation[43,54,88]. After 8.3 ns, a fourth species is formed and does not display the signatures of (S₁S₀) (Fig. 5e and Supplementary Fig. 21). Importantly, the fourth and fifth species display the same spectral properties as that of (T₁) with lifetimes of 29.7 ns and 77.5 μs, respectively. Thus, the fourth and fifth species are assigned as the correlated and uncorrelated triplet pair states, (T₁T₁) and (T₁ + T₁), respectively. The triplet quantum yields of mPh(Tc-Py)₂ are approximated by singlet oxygen quantum yield (Φ_Δ) measurements with a value of 113% in

toluene. It is important to note that Φ_Δ does not represent the yield of a pure free triplet excited state, since both (T₁T₁) and mixed states are likely to be quenched by O₂[89–91]. The dissociation yield (Φ_Diss) of the triplet pair state was determined by calculating the ratio of ΔOD values corresponding to the evolution-associated spectra of (T₁T₁) and (T₁ + T₁). Accordingly, Φ_Diss of mPh(Tc-Py)₂ in toluene is low, ca. 8%, which is similar to previous reports for meta-phenylene-linked tetracene and pentacene dimers[43,54]. The strong electronic coupling between the two tetracenes moieties through the meta-phenylene spacer enhances exchange interactions between ¹(T₁T₁) and ⁵(T₁T₁), consequently suppresses spin-mixing of ¹(T₁T₁)-⁵(T₁T₁), which ultimately renders dissociation of (T₁T₁) highly unlikely[19,40,41,92,93]. Moreover, the evolution of triplet excited state for mPh(Tc-Py)₂ does not depend on the photoexcitation wavelength (Supplementary Fig. 22), suggesting a rigid and structurally homogeneous conformation in solution, consistent with the observation in steady state absorption analysis.

Examining data from TCSPC measurements in toluene reveals a tri-exponential decay with a prompt decay of faster than 200 ps and two slower decays of 7.4 and 27.4 ns (Supplementary Fig. 26a and Supplementary Table 4). Notably, these lifetimes match well those of (S₁S₀)ᵣₑₗ, (M), and (T₁T₁), respectively. In time-resolved emission spectroscopy (TRES) studies, the analogous fluorescence signatures that were seen in the steady-state fluorescence measurements (maxima at 580 and 625 nm) dominate throughout the entire deactivation process (Supplementary Fig. 27a–f). Deconvolution of the raw data from TRES using a three-species sequential kinetic model yields three effectively identical fluorescence spectra. In other words, the prompt fluorescence stems directly from (S₁S₀)ᵣₑₗ, while the two delayed fluorescence

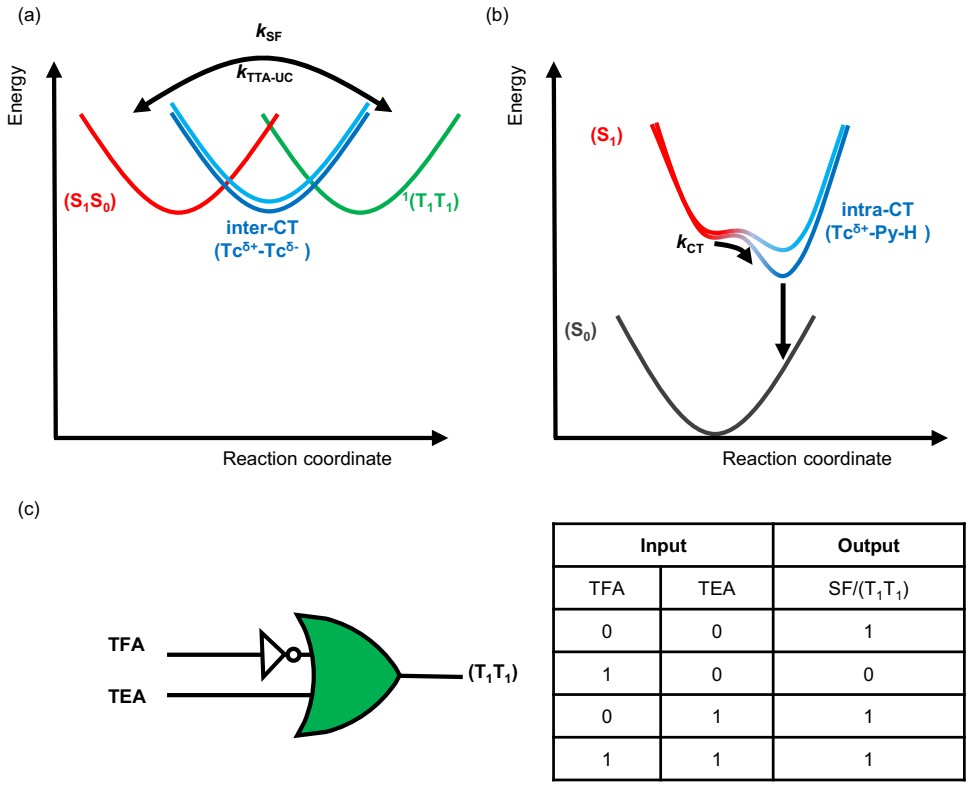

**Fig. 7 | Summary of switchable SF for mPh(Tc-Py)$_2$. a** Proposed schematic potential energy surface of mPh(Tc-Py)$_2$, in which $(S_1S_0)$ is in red, inter-CT state is in blue (dark blue represents inter-CT state in benzonitrile and sky blue represents inter-CT state in toluene), and $^1(T_1T_1)$ is in green. **b** Proposed schematic potential energy surface of Ph(Tc-Py-H$^+$) and mPh(Tc-Py-H$^+$)$_2$ in which $(S_1)$ is in red, intra-CT state is in blue (dark blue represents intra-CT state in benzonitrile and sky blue represents intra-CT state in toluene), and $(S_0)$ is in black (**c**) Schematic diagram of the IMPLICATION logic gate operation with TFA and TEA as the inputs and singlet fission or the formation of $^1(T_1T_1)$ as outputs the corresponding truth table.

events arise from recombination involving (M) and (T$_1$T$_1$), respectively. It is notable that the dynamics of mPh(Tc-Py)$_2$ in toluene are similar to our previous investigations with mPhTc$_2$[43].

To confirm that an inter-CT state mediates SF, we then examined the dynamics of mPh(Tc-Py)$_2$ in the more polar solvent benzonitrile, in which CT becomes energetically more accessible. The fs- and ns-TAS data are shown in Fig. 6 and Supplementary Figs. 23d–f and 24. The initial dynamics in benzonitrile are comparable to those in toluene. In particular, we note the coexistence of $(S_1S_0)_{rel}$ and $(T_1T_1)$ that derive from $(S_1S_0)$. Interestingly, after hundreds of picoseconds, a new ESA appears in the range of 800–1200 nm, which is reminiscent of the signature of the one-electron oxidized form of tetracene (Supplementary Fig. 23d–f)[43,79,80]. At later times, the coexistence of $(S_1S_0)_{rel}$, CT, and (T$_1$T$_1$) states continues for several hundred nanoseconds. The fs- and ns-TAS data in benzonitrile are best fit by a four-species sequential model (Fig. 6; Supplementary Figs. 23d–f and 25b; Supplementary Table 3). EAS of the first species has the characteristics of (S$_1$) for Ph(Tc-Py) in benzonitrile and is, therefore, ascribed to $(S_1S_0)$. After 21.7 ps, the second species is formed, and ESA reflects that of (T$_1$T$_1$) at around 470 nm and that of $(S_1S_0)_{rel}$ in the range from 600 to 800 nm. In other words, this species reflects an initial mixed state (M$_1$), which is a combination of the $(S_1S_0)_{rel}$, CT, and (T$_1$T$_1$) states. Notably, (M$_1$) is similar to (M) in toluene, in contributions from the CT state are mostly masked by contributions from $(S_1S_0)$. In benzonitrile, (M$_1$) is significantly shorter-lived (220 ps) than (M) in toluene (8.3 ns). The next, third, species shows EAS in the visible range similar to that of (M$_1$), but the contributions at 470 nm from (T$_1$T$_1$) are somewhat reduced. At the same time, the signature at 800 to 1200 nm, which relates to the one-electron oxidized form of tetracene, is enhanced[43,79,80]. The third species is still a mixed state, namely (M$_2$), in which (T$_1$T$_1$) contributions are slightly weakened, those of the CT state became much stronger, and

those of $(S_1S_0)$ remained unchanged. Subsequently, (M$_2$) converts to the fourth species after ca. 12.9 ns. The fourth species displayed weak EAS intensities, but the signature of (T$_1$) and the lifetime (94.5 µs) are unmistakable – it is the uncorrelated triplet excited state (T$_1$ + T$_1$). Interestingly, the mixed state (M$_2$) of mPh(Tc-Py)$_2$ in benzonitrile failed to terminate to a pure (T$_1$T$_1$) state as observed in toluene. A likely explanation is that mixing with the energetically low-lying inter-CT state enhances the coupling between $(S_1S_0)$ and (T$_1$T$_1$), which hampers the evolution of a pure state. The singlet oxygen quantum yield $\Phi_\Delta$ in benzonitrile is only 82%. We note that the fluorescence of mPh(Tc-Py)$_2$ is mono-exponential in benzonitrile and that the lifetime, 13.6 ns, is similar to that of (M$_2$) (Supplementary Fig. 26b and Supplementary Table 3). TRES reveals that the sole fluorescent state is identical to the steady-state fluorescence spectrum in benzonitrile (Supplementary Fig. S27d–f). Therefore, the fluorescent state is $(S_1S_0)_{rel}$, and the delayed contribution comes from (M$_2$). We postulate that the ultrafast formation of (T$_1$T$_1$), as well as the rapid evolution of (M$_1$), restricts prompt fluorescence from the initial state $(S_1S_0)_{rel}$, and the delayed fluorescence occurs from (M$_1$). Overall, the weak fluorescence from these two pathways renders the prompt and the delayed fluorescence non-detectable via TCSPC. In summary, inter-CT state gates the formation of (T$_1$T$_1$) in mPh(Tc-Py)$_2$, but does not inhibit SF regardless of the solvent, as depicted in Fig. 7a.

After protonation, mPh(Tc-Py-H$^+$)$_2$ exhibits the same excited-state dynamics as Ph(Tc-Py-H$^+$), and, in particular, the formation of triplet excited states is absent, regardless of the solvent or photoexcitation wavelength (Supplementary Figs. S28–30). Thus, SF is successfully deactivated in mPh(Tc-Py)$_2$ by the acid stimuli, and, alternatively, an intra-chromophore charge transfer between tetracene and pyridinium governs the excited-state dynamics. Upon protonation of mPh(Tc-Py)$_2$, the intra-CT state assumes the role of a

trap state that outcompetes $(T_1T_1)$ formation (Fig. 7b). To establish the reversibility of the acid-base switching of SF for mPh(Tc-Py)$_2$, fs-TAS of mPh(Tc-Py)$_2$ was recorded in toluene upon the sequential addition of TFA and TEA (Supplementary Fig. 31). In fs-TAS analysis of mPh(Tc-Py)$_2$ after the sequential addition of TFA and TEA, the characteristics of $(T_1T_1)$ at 470 nm are observed within hundreds picoseconds along with the characteristics of $(S_1S_0)_{rel}$ in the range of 600–750 nm. In contrast to the complete reversibility found in the steady-state absorption spectra of mPh(Tc-Py)$_2$ via the sequential switching with TFA-TEA in toluene, mPh(Tc-Py)$_2$ exhibits slight changes in the fs-TAS in toluene. From fs-TAS, it is shown, for example, that contributions from $(S_1S_0)_{rel}$ are increased in the mixed state (species 3 in Supplementary Fig. 31), likely due to the increased polarity of solvent from the increasing concentration of salts from acid and base inputs. Thus, the complete resetting of the environment is hindered by salt accumulation, which affects the perfect reversibility of SF switches.

The switchable, reversible SF behavior of mPh(Tc-Py)$_2$ toward TFA and TEA encouraged us to consider its application as an IMPLICATION gate using chemical input (TFA/TEA) (Fig. 7c). IMPLICATION logic is the combination of NOT and OR gates. Applying this logic gate to mPh(Tc-Py)$_2$, TFA and TEA provide two Boolean inputs. The output '1' represents SF ON with the multiexciton triplet pair as the probe, whereas output '0' represents SF OFF with a multiexciton triplet pair that is not formed. As shown in the logic truth table, in the absence of TFA or the presence of both TFA and TEA, $(T_1T_1)$ is the probe, and the output is 1 when the input signals are (0,0), (0,1), and (1,1). Adding TFA, (1,0) as the input generates the output 0, that is, SF is inhibited and $(T_1T_1)$ is not formed. Next to TAS, emission spectra of mPh(Tc-Py)$_2$ also serve as sensitive probe for the reversibility of SF. A broad intra-CT emission dominates from mPh(Tc-Py-H$^+$)$_2$ and signifies the inhibition of SF (SF OFF). In stark contrast, local excited-state emission of mPh(Tc-Py)$_2$ indicates SF ON.

## Discussion

In conclusion, we have successfully developed a reversible, external stimulus-responsive system for SF based on the combination of a tetracene dimer featuring a *meta*-phenylene-spacer and pendent pyridyl groups. Our strategy centers on the reversible tuning of the high-energy inter-CT and low-energy intra-CT states using acid as an external stimulus and, in turn, realizing a means to turn singlet fission 'on-off' through protonation of the pyridyl moieties to give mPh(Tc-Py-H$^+$)$_2$. Without acid stimuli, a mixed state stemming from the superposition of the $(S_1S_0)_{rel}$, inter-CT, and $(T_1T_1)$ states evolves as an intermediate that promotes SF in mPh(Tc-Py)$_2$. The treatment of mPh(Tc-Py)$_2$ with TFA results in the formation of the corresponding pyridiniums groups, which are stronger electron acceptors than the free-base dimer. Consequently, an energetically lower-lying intra-CT state acts as a trap state in the protonated dimer, and, in this way, SF is precluded. Finally, the duality and switchability of the mPh(Tc-Py)$_2$/mPh(Tc-Py-H$^+$)$_2$ system is considered as an IMPLICATION logic gate using TFA and TEA as inputs and formation of the correlated triplet pair state as an output. To the best of our knowledge, this is the first case of a switchable triplet pair via SF. In this project, we utilized acid and base as stimuli to realize the triplet pair/SF switch and molecular logic gate. Beyond this, other inputs, such as optical signals, electronic signals, and magnetic signals, hold significant potential for the development of more diverse SF switches. Notably, future investigations could address not only the 'on' and 'off' of SF, but also the controllable multiplicities of the triplet pair. Furthermore, the integration of multiple input signals could enable the design of more complex molecular logic gates based on SF. This work not only validates the viability of switchable multiple-exciton generation through a modulating CT state, but also opens the way for broader developing stimulus-responsive SF materials.

## Methods

Reagents were purchased reagent grade from commercial suppliers and used without further purification. Dry tetrahydrofuran was obtained from a commercial solvent purification system (LC Technology Solutions INC). MgSO$_4$ was used as the drying reagent after aqueous work-up.

$^1$H and $^{13}$C NMR spectra were recorded on an Agilent/Varian Inova four-channel 500 MHz spectrometer ($^1$H: 498 MHz), or an Agilent/Varian VNMRS two-channel 500 MHz spectrometer equipped with a $^{13}$C/$^1$H dual cold probe ($^1$H: 500 MHz, $^{13}$C: 126 MHz). NMR spectra were recorded at ambient probe temperature and referenced to the residual solvent signal ($^1$H: CDCl$_3$: 7.26 ppm, $^{13}$C: CDCl$_3$: 77.06 ppm). The coupling constants of protons in all $^1$H spectra have been reported as pseudo first-order when possible, even though they can be higher-order (ABC, ABX, AA'BB') spin systems; coupling constants are reported as observed. High resolution mass spectra were obtained from a Bruker 9.4 T Apex-Qe FTICR instrument. IR spectra were recorded on a Thermo Nicolet 8700 FTIR spectrometer and continuum FTIR microscope as films. Differential scanning calorimetry (DSC) measurements were made on a Mettler Toledo DSC or Perkin Elmer Pyris 1 DSC. All DSC measurements were carried out under a flow of nitrogen with a heating rate of 10 °C/min. Melting points were measured with 6406-K Thomas-Hoover melting point apparatus with periscopic thermometer reader. Thin layer chromatography (TLC) analyses were carried out on TLC glass plates from Merck KGaA and visualized via UV-light (254/364 nm). Column chromatography used Supelco™ silica gel (SiO$_2$, 60 Å, 40–63 µm).

Steady-state UV-vis absorption measurements were carried out with a UV-1900i (Shimadzu) two-beam spectrophotometer, and steady-state fluorescence measurements were carried out using an Edinburgh FS5 spectrometer. For molar extinction coefficients, repeated measurements verified an error in the range of ±8%. The fluorescence quantum yields were determined via an absolute method using an integrating sphere setup, and the uncertainties of 5–10% can be achieved as reported[94]. Here, the error of ±10% for absolute fluorescence quantum yield is considered.

Time-correlated single-photon counting (TCSPC) lifetimes and time-resolved fluorescence spectra (TRES) were measured by a Fluorolog 3 time-correlated photon counting instrument from Horiba Jovin Yvon, a SuperK Fianium FIU-6PP supercontinuum laser from NKT Photonics as the excitation source, and an R3809U-50 MCP photomultiplier from Hamamatsu. A Horiba Jobin Yvon FluoroLog3 emission spectrometer with a Symphony II detector in the near-infrared (NIR) detection range recorded the singlet oxygen emission spectra. The singlet oxygen quantum yield is determined by using C60 as a standard. All spectra were acquired at room temperature using 10 × 10 mm quartz glass cuvettes.

Ultrafast pump-probe transient absorption spectroscopy (TAS) was performed using an Astrella-F-1K amplified Ti:sapphire femtosecond laser system from Coherent, operating at a repetition rate 1 kHz, 5.5 W power (5 mJ pulse energy), pulse duration of 80 fs. To acquire the time resolved transient absorption spectra on a sub-ps and ns resolution, an Ultrafast Systems HELIOS or EOS fs/ns transient absorption spectrometer was used with time delays from 0 to 7200 ps and 1 ns to 400 µs, respectively. For sub-ps, white light for the probing pulse in the visible region of the optical spectrum (~420–770 nm) was generated by focusing part of the fundamental 800 nm output onto a 2 mm sapphire disk. For (near) IR (800–1350 nm) white light, a 10 mm sapphire was used. For ns timescale experiments, white light for probing was generated by a photonic crystal fiber supercontiuum laser with a 1064 nm fundamental. The excitation wavelength was generated via a TOPAS Prime from Light Conversion with standard NirUVis extension. Data evaluation of the fs-TAS, ns-TAS, and TRES data was conducted by a combination of multiwavelength and global analysis using the GloTarAn software, which is a free, Java-based graphical user interface to

the R-package TIMP[95–97]. For parameters determined via transient absorption measurements, an error in the range of ±10% is considered, as this is the typical error in photophysical measurements.

The molecular structures of the investigated molecules were optimized by the Gaussian 16 software package[98] on the B3LYP-GD3BJ[99]/def2SVP[100] level in the gas phase.

## Data availability
The authors declare that all data supporting this study are available in the main manuscript or in the Supplementary Information. The single crystal data of Ph(Tc-Py) can be obtained in the Cambridge Structural Database corresponding to CCDC number 2369785. The Source data in this study have been deposited in the FigShare under accession code https://figshare.com/s/e85057574bc6cb4650c0. Source data are provided with this paper.

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

## Acknowledgements

R.R.T. is grateful for funding from the Natural Sciences and Engineering Research of Canada (NSERC, grant no. RGPIN-2017-05052) [R.R.T.] and the Canada Foundation for Innovation (CFI). D.M.G. acknowledges financial support from the Deutsche Forschungsgemeinschaft (DFG) as part of SFB 953 "Synthetic Carbon Allotropes", GU 517/32-1 [D.M.G.] and GU 517/27-1 [D.M.G.]. Y.B. acknowledges a fellowship from the Chinese Scholarship Council.

## Author contributions

Y.B., Y.H., R.R.T., and D.M.G. designed the research. Y.B. performed the photophysical studies. Y.H. and D.A.X.L. performed the synthesis. Y.B. and T.C. conducted the theoretical calculations. Y.B., Y.H., R.R.T., and D.M.G. analyzed the data. M.J.F. performed the single crystal diffraction studies. Y.B., Y.H., R.R.T., and D.M.G. wrote the paper with contributions from all authors.

## Funding

## Competing interests

The authors declare no competing interests.
