## [Transparent Peer Review file · Nature Communications]

Reversible gating of singlet fission by tuning the role of a charge-transfer state

Corresponding Author: Professor Dirk M. Guldi

Version 0:

Reviewer comments:

Reviewer #1

(Remarks to the Author)

The manuscript by Bo et al., details the controllable singlet fission switching behavior in a meta-diethynylphenylene-linked tetracene dimer with pyridyl end groups. The authors use a combination of time-resolved and steady-state absorption and emission spectroscopies to demonstrate that the tetracene dimer undergoes efficient singlet fission in solution, both in toluene and benzonitrile. The higher polarity solvent enhances the involvement of a CT state in the dynamics, but not enough to act as trap to inhibit singlet fission. However, upon protonation of the pyridyl end groups to form an electron-poor pyridinium unit, a new charge transfer state is formed that is significantly lower in energy, trapping by which then kinetically outcompetes singlet fission and shuts off triplet formation. The authors demonstrate that this process is reversible over many cycles by successive application of chemical reducing and oxidizing agents, thus demonstrating stimuli-responsive singlet fission. The strategy is simple but clever, the experiments are convincing, and the manuscript is clearly written. This study will be of significant interest to the community, which has not seen many examples of controllable singlet fission in these types of systems. However, there are some issues of clarity that should be addressed, as discussed below, prior to publication in Nature Communications.

Major issues:

-The authors show convincing evidence that singlet fission in the dimer in its pyridyl form is mediated by the CT state, specifically the (tetracene⁺/tetracene⁻) CT state. They then show that in the pyridinium form, the charge transfer state inhibits SF. It is implied cursorily that these two CT states are the same; Figure 6 shows a single "(CT)" potential energy surface. However, these states are not the same, since the charge transfer in the protonated case is (tetracene⁺/pyridinium⁰) in nature. Indeed, in their transient spectroscopic assignments the authors do state which charged moieties contribute to which spectral features, and they are different between the cases. This makes Figure 6 somewhat misleading and some of the discussion unclear. The identity of the two different charge transfer states should be stressed and clarified throughout the manuscript, and Figure 6 revised accordingly.

-The authors should clarify why the fluorescence quantum yield increases for Ph(Tc-Py) and mPh(Tc-Py)₂ in benzonitrile compared to toluene, particularly in the monomer since there is no ability there for state mixing. The shorter lifetime of the monomer in benzonitrile compared to toluene (7.9 compared to 8.2 ns with no error bars, see below) combined with the higher quantum yield implies that the non-radiative internal conversion rate is changing dramatically. The authors should comment on this to better establish the photophysics of the isolated molecule.

-Similarly, why is the charge transfer time in Ph(Tc-Py-H⁺) slower in benzonitrile than in toluene, despite the faster decay of the emissive CT state, as expected? For example, is this because of solvent gating or an energy (Marcus) effect?

-It would be useful if the authors can give a little more discussion about the yields of the various triplet states. The experimental estimations of the T₁+T₁ yields from oxygen sensitization are quite helpful as lower bounds, but can they be correlated with estimates of the yields from kinetic analysis? This might be difficult given the mixed nature of the S₁S₀/CT/TT state, but that intermediate appears to be formed in very high yield. The experimental estimates for the T₁+T₁ yields would then seem to imply significant spin evolution prior to decorrelation, potentially involving the 3TT state that diminishes the overall uncorrelated triplet yield. Without direct probing of these effects by spin-sensitive techniques such discussion will admittedly be speculative, but it may point to loss mechanisms that could be considered in future designs.

-Are the TRES component spectra observed for mPh(Tc-Py)₂ identical, or are there subtle shifts that might imply alterations in the state energies during the mixing process? Normalization of the component spectra in Figure S22b will address this.

-The IMPLICATION gate is claimed to be an "all-optical" logic gate (line 369). However, this seems odd because only the readout in this case is optical; the actions themselves are chemical. At the very least this should be clarified, but that

particular claim seems like a stretch and should be downplayed.

-Error bars/uncertainties should be presented for all values in the main text, and in the tables in the Supplementary Information.

-A kinetic scheme for the models used and the energies of the states in the dimer would be useful, even in the Supplementary Information.

Minor issues:

-Line 210: "The kinetic model of the ns-TAS data..." The sentence goes on to discuss time constants on the order of \sim ps, indicating that this is actually the "fs-TAS" data.

-Line 361 is confusingly worded, as it seems to imply that the deprotonation is being monitored on the ultrafast timescale. Please revisit.

-There are several typos in the SI: Figures S14 and S16 both say they present TAS data for the NIR region despite showing data for the visible spectrum.

Reviewer #2

(Remarks to the Author)

Bo et al. report on the singlet fission behavior of a tetracene dimer and the response of the photophysics to protonation/deprotonation of pendant amine groups. The authors show that protonation results in a significant enhancement in charge-transfer interactions, leading to a rapid CT-based decay pathway that outcompetes singlet fission. After deprotonation, singlet fission behavior (previously demonstrated in a related dimer) is restored. The basic spectroscopic behavior is clearly demonstrated, and the results are largely well explained. The study certainly merits publication, though I am not convinced that the novelty/significance rises to the level of Nature Communications: the basic concept is that chemical tuning that distorts the electronic structure will disrupt the photophysics (in this case singlet fission). It is very clearly demonstrated, but not particularly surprising. I have no major criticisms of the technical aspects of the work, however, which seem to be at a good level. Below I list some minor points that should be addressed prior to publication, for clarification or context.

1. The authors should provide normalized EAS for comparison in the SI. For instance, in line 271-272 they note that the fourth species 'does not display the signatures of (S1S0)'. From the text or figure it is not immediately clear which signatures these are (what wavelengths/peaks are key to this analysis?), while the intensity differences between different EAS make it extremely difficult to evaluate what, if anything distinguishes the species.

2. In the global analysis of the excited-state dynamics, the authors require a large number of components (4 or 5 depending on solvent) to accurately represent the data. This appears to be because of the choice of a sequential model. Some of the authors, as well as others in the field (e.g. Ringstrom et al., DOI: 10.1039/D1SC06285A and Kim et al., DOI: 10.1016/j.xcrp.2024.102045), have shown that such acetylene-bridged acene dimers inevitably exhibit significant rotational disorder. Is a possible alternative explanation of the reported dynamics of the variously singlet-CT-TT mixed states that they are capturing the evolution of a disordered ensemble, effectively a branched/parallel decay pathway?

3. The authors describe the delayed emission once the system has evolved into the 1(T1T1) state as an upconversion process. But in Figure 6 the 1(T1T1) state is drawn distinctly higher in energy than S1S0, which would suggest a downconversion/relaxation. The description in the text and the schematic PES should be brought into agreement.

4. Have the authors investigated a pump wavelength dependence? A previous quite relevant study on intramolecular singlet fission in a tetracene dimer seems to have been missed: Alvertis et al., DOI: 10.1021/jacs.9b05561. That work also invoked the idea of a 'switch', where the role and energy of charge-transfer states was a critical factor. In particular, the regime where a CT state served as a 'trap', as reported here in the TFA-treated dimer, could be apparently overcome by excitation at higher energy. It would be interesting to know if the logic behavior schematically portrayed in Figure 6 changes with excitation conditions.

5. There is an extra Ph(Tc-Py-H+) on line 227.

Reviewer #3

(Remarks to the Author)

The authors describe an acid/base stimuli-responsive conjugate for singlet fission. They show convincingly that charge-transfer states have a direct impact on the occurrence of singlet fission. While the non-protonated conjugate shows singlet fission, protonation leads to an energetically stabilized charge-transfer state that acts as a sink thereby avoiding singlet fission.

The photophysical description of the involved pathways and transients is nicely supported by in-depth steady-state and time-resolved spectroscopic experiments.

The stimuli-responsiveness is summarized as a binary logical operation (IMP gate). This one I would not call "an all-optical" gate, because the input signals are not optical signals.

The work indeed opens new avenues for the design of switchable singlet-fission materials. Conceptually the use of acid and base is fine and the performance for a reduced number of switching cycles seems reasonable. However, in the long term the repeated acid-base neutralization will accumulate salt waste, which may influence the overall polarity of the medium (which ultimately would have impact of the singlet fission efficiency). Maybe this could be mentioned in the paper.

The subtle interplay between charge-transfer states as "mediators or inhibitors" of the singlet-fission process is a nice idea and for providing an experimental demonstration of this idea the manuscript is a strong candidate for publication in Nat. Commun.

The translation of the experimental switching into an IMP logic is correct, but from my point of view it does not add too much

to the paper. The herein described logic feature is comparably simple. Molecular logic (with chemical, optical or other inputs) has simulated much more complex logic operations in the past. Therefore, I would advise to base the introductory focus on the conceptual novelty of the switching process itself rather than on the resemblance with logic operations.

Version 1:

Reviewer comments:

Reviewer #1

(Remarks to the Author)

The authors have satisfactorily addressed my original concerns and I believe the revised scope and emphasis of the article make it suitable for publication in Nature Communications.

Reviewer #2

(Remarks to the Author)

The authors have largely satisfied my concerns – the additional context and improved analysis are helpful. My only remaining concern is a minor one regarding the spectral decomposition. In the dimer, the authors invoke a state M in Figure 4 with a mixture of S1, T1T1, and inter-CT character. While that is plausible in principle, in the accompanying text the authors only describe spectral features related to S1 and T1T1. From close inspection of the figures, I concur that there is no identifiable CT character (it should manifest, for instance, as a difference in spectral shape between M and the final triplet state in the normalized spectra in the SI). Given there is no evidence for any involvement of CT character in the state M, I do not see how the authors can justify invoking it, and its mention should be removed. The authors' papers are widely read in the field, and it sets a dangerous precedent to make such assignments without direct support in the spectra.

Reviewer #3

(Remarks to the Author)

The authors have presented a revised version of their manuscript, in which they have addressed my minor criticism in full. As far as I can see they have done also a very good job in revising their work regarding open points of their photophysical discussion (as suggested by my reviewer colleagues). From my point of view this insightful manuscript could be now accepted for publication.

Response to Reviewers

Dear Reviewers:

Thank you for your careful review and valuable suggestions. We have addressed all comments by revising the Manuscript and the Supplementary Information and provide a point-by-point response listed below. The changes in the revised version of the manuscript and the supporting information are highlighted in yellow, and the clarifications regarding the comments are provided in red.

Thank you very much for your kind consideration.

Dirk M. Guldi

Professor of Chemistry

REVIEWER COMMENTS

Reviewer #1 (Remarks to the Author):

The manuscript by Bo et al., details the controllable singlet fission switching behavior in a meta-diethynylphenylene-linked tetracene dimer with pyridyl end groups. The authors use a combination of time-resolved and steady-state absorption and emission spectroscopies to demonstrate that the tetracene dimer undergoes efficient singlet fission in solution, both in toluene and benzonitrile. The higher polarity solvent enhances the involvement of a CT state in the dynamics, but not enough to act as trap to inhibit singlet fission. However, upon protonation of the pyridyl end groups to form an electron-poor pyridinium unit, a new charge transfer state is formed that is significantly lower in energy, trapping by which then kinetically outcompetes singlet fission and shuts off triplet formation. The authors demonstrate that this process is reversible over many cycles by successive application of chemical reducing and oxidizing agents, thus demonstrating stimuli-responsive singlet fission. The strategy is simple but clever, the experiments are convincing, and the manuscript is clearly written. This study will be of significant interest to the community, which has not seen many examples of controllable singlet fission in these types of systems. However, there are some issues of clarity that should be addressed, as discussed below, prior to publication in Nature Communications.

Author reply: We appreciate the reviewer's professional comments and kind recommendation of our work. We do hope that our point-to-point response below will clear up their concerns.

Major issues:

Reviewer query: The authors show convincing evidence that singlet fission in the dimer in its pyridyl form is mediated by the CT state, specifically the (tetracene⁺/tetracene⁻) CT state. They then show that in the pyridinium form, the charge transfer state inhibits SF. It is implied cursorily that these two CT states are the same; Figure 6 shows a single "(CT)" potential energy surface. However, these states are not the same, since the charge transfer in the protonated case is (tetracene⁺/pyridinium⁰) in nature. Indeed, in their transient spectroscopic assignments the authors do state which charged moieties contribute to which spectral features, and they are different between the cases. This makes Figure 6 somewhat misleading and some of the discussion unclear. The identity of the two different charge transfer states should be stressed and clarified throughout the manuscript, and Figure 6 revised accordingly.

Author reply: We thank the reviewer for these comments and apologize for the vague description. First, to distinguish between these two different charge states, we have renamed them. In the revised version of the manuscript, we have used the inter-chromophore charge transfer state (inter-CT) to represent (Tc^{δ+}-Tc^{δ-}), which mediates singlet fission. The intra-

chromophore charge transfer (intra-CT) represents ($\text{Tc}^{\delta+}\text{-Py-H}$), which is the trap state that is present in the protonated form of the monomer and dimer. Across the revised version of the manuscript, the difference between these two states has been marked to improve clarity and avoid any further confusion. Moreover, the Figure 6 has been revised accordingly.

Please see the revised version of the manuscript: lines 216–218 highlighted in yellow.

The underlying nature, that is, an **intra-chromophore CT (intra-CT)** from tetracene to pyridinium, is corroborated by a fluorescence that is sensitive to the solvent polarity (Table S1; Figures 2b and S12).

Please see the revised version of the manuscript: lines 289–291 highlighted in yellow.

As such, we assign this second species as an **intra-chromophore CT (intra-CT)** state, representing the charge transfer between pyridinium and tetracene ($\text{Tc}^{\delta+}\text{-Py-H}$).

Please see the revised version of the manuscript: lines 336–338 highlighted in yellow.

Here, the CT state is the charge transfer between two tetracenes; we ascribe it to the **interchromophore CT (inter-CT)** state.

Please see the revised version of the manuscript: Figures 6a and 6b highlighted in yellow.

Figure 6. (a) Proposed schematic potential energy surface of **mPh(Tc-Py)₂**, in which (S_1S_0) is in red, inter-CT state is in blue (dark blue represents inter-CT state in benzonitrile and sky blue represents inter-CT state in toluene), and ${}^1(T_1T_1)$ is in green. (b) Proposed schematic potential energy surface of **Ph(Tc-Py-H⁺)** and **mPh(Tc-Py-H⁺)₂** in which (S_1) is in red, intra-CT state is in blue (dark blue represents intra-CT state in benzonitrile and sky blue represents intra-CT state in toluene), and (S_0) is in black

Reviewer query: The authors should clarify why the fluorescence quantum yield increases for Ph(Tc-Py) and mPh(Tc-Py)₂ in benzonitrile compared to toluene, particularly in the monomer since there is no ability there for state mixing. The shorter lifetime of the monomer in benzonitrile compared to toluene (7.9 compared to 8.2 ns with no error bars, see below) combined with the higher quantum yield implies that the non-radiative internal conversion rate is changing dramatically. The authors should comment on this to better establish the photophysics of the isolated molecule.

Author reply: We greatly appreciate this valuable comment. Based on this suggestion, we have determined the radiative and non-radiative decay rate constants of monomer **Ph(Tc-Py)** to gain more insights into the photophysics. First, **Ph(Tc-Py)** exhibits similar radiative decay rate constants regardless of the solvent; 9.7×10^7 and 11.0×10^7 s⁻¹ in toluene and benzonitrile, respectively. These radiative decay rate constants (k_r) are similar to those seen for unsubstituted tetracene and tetracene derivatives: $3\text{--}10 \times 10^7$ s⁻¹. (10.1016/S1386-1425(97)00071-1; 10.1016/0584-8539(88)80084-9). However, the nonradiative decay rate constants (k_{nr}) of **Ph(Tc-Py)** [2.4×10^7 s⁻¹ in toluene and 1.4×10^7 s⁻¹ in benzonitrile] are smaller by an order of magnitude than that observed in **unsubstituted tetracene** [ca. $1\text{--}3 \times 10^8$ s⁻¹ (10.1016/S1386-1425(97)00071-1; 10.1016/0584-8539(88)80084-9)]. The decrease in k_{nr} is due to the absence of intersystem crossing in **Ph(Tc-Py)**, whose yield exceeds 0.62 in tetracene (10.1016/0584-8539(88)80084-9). In turn, **Ph(Tc-Py)** displays higher fluorescence quantum yields, regardless of the solvent, especially when compared to tetracene. Thus, k_{nr} for **Ph(Tc-Py)** is the rate constant of internal conversion (k_{IC}). It is noted that k_{IC} of **Ph(Tc-Py)** is solvent dependent and is higher in toluene than in benzonitrile. First, k_{IC} of **Ph(Tc-Py)** in different solvents has the same magnitude of 10^7 s⁻¹ as **unsubstituted tetracene**, and the difference between k_{IC} of **Ph(Tc-Py)** in different solvents thus stems from solvent effects. In more viscous benzonitrile, intramolecular rotation is hindered in **Ph(Tc-Py)**, leading to a reduction in the non-radiative decay, a smaller k_{nr} in benzonitrile, and ultimately a slightly higher fluorescence quantum yield. Similar phenomenon is observed in other fluorophores (10.1021/ja800570d; 10.1016/1010-6030(89)87093-5; 10.1038/nchem.120). We have added this discussion in the revised version of the manuscript to better establish the photophysics of monomer **Ph(Tc-Py)**.

The details of determination of k_r and k_{nr} have been elaborated in the following:

The fluorescence quantum yield (Φ_F) is the number of photons emitted per photons absorbed by the system:

$$\Phi_F = \frac{\text{number of photons emitted}}{\text{number of photons absorbed}} \quad (\text{Equation R1})$$

It has been possible to express Φ_F in terms of the rates of radiative and non-radiative decays. Thus, Φ_F has been determined by the balance between radiative and non-radiative rate constants:

$$\Phi_F = \frac{k_r}{k_r + \sum k_{nr}} \quad (\text{Equation R2})$$

Here, k_r is the radiative transition rate, whereas Σk_{nr} is the sum of all non-radiative decay rates, including processes such as internal conversion (k_{IC}) and intersystem crossing (k_{ISC}). The lifetime of our emissive state has been represented by one over the sum of the rate constants:

$$\tau_F = \frac{1}{k_r + \Sigma k_{nr}} \quad (\text{Equation R3})$$

In turn, using Equation R2 and R3, k_r has been calculated according to:

$$k_r = \frac{\Phi_F}{\tau_F} \quad (\text{Equation R3})$$

Please see the revised version of the manuscript: lines 256–271 highlighted in yellow.

From the fluorescence quantum yields and lifetimes of $(S_1)_{rel}$, the radiative (k_r) and nonradiative (k_{nr}) rate constants of **Ph(Tc-Py)** in different solvents are determined (Table S2). Similar to unsubstituted tetracene (**Tc**) and other tetracene monomers with k_r in the range of $3\text{--}10 \times 10^7 \text{ s}^{-1}$,^{74,75} **Ph(Tc-Py)** exhibits radiative k_r of 9.7×10^7 and $11.0 \times 10^7 \text{ s}^{-1}$ in toluene and benzonitrile, respectively. Constants k_{nr} of **Ph(Tc-Py)** are 2.4×10^7 and $1.4 \times 10^7 \text{ s}^{-1}$ in toluene and benzonitrile, respectively, which are one order of magnitude lower than that of unsubstituted tetracene ($1\text{--}3 \times 10^8 \text{ s}^{-1}$).^{74,75} Changes in k_{nr} are attributed to the absence of intersystem crossing (ISC) in **Ph(Tc-Py)**, where the yields of ISC of **Tc** are larger than 0.62.⁷⁵ As a result, **Ph(Tc-Py)** shows stronger fluorescence than **Tc** regardless of the solvent. For **Ph(Tc-Py)**, k_{nr} is dominated by internal conversion (k_{IC}). Notably, k_{IC} of **Ph(Tc-Py)** varies as a function of solvents and is higher in toluene than in benzonitrile, but remains around 10^7 s^{-1} , similar to k_{IC} of **Tc**. Differences in terms of k_{nr} of **Ph(Tc-Py)** in toluene ($2.4 \times 10^7 \text{ s}^{-1}$) and benzonitrile ($1.4 \times 10^7 \text{ s}^{-1}$) likely stem from solvent effects. In a more viscous benzonitrile, the restriction of intramolecular rotation reduces k_{IC} (k_{nr}) and, thereby increases the fluorescence quantum yields. This phenomenon is consistent with observations made for other fluorescent materials.^{76–78}

Please see the revised version of the supporting information: Table S2 highlighted in yellow.

Table R1 (Table S2). Summary of photoluminescence behavior of **Ph(Tc-Py)** in toluene and benzonitrile.

Ph(Tc-Py)	Φ_F	$\tau (S_1)_{rel}$ (ns)	$k_r (s^{-1})$	$k_{nr} (s^{-1})$
toluene	0.80 ($\pm 10\%$)	8.2 ($\pm 10\%$)	9.7×10^7 ($\pm 14\%$)	2.4×10^7 ($\pm 14\%$)
benzonitrile	0.89 ($\pm 10\%$)	7.9 ($\pm 10\%$)	11.0×10^7 ($\pm 14\%$)	1.4×10^7 ($\pm 14\%$)

Reviewer query: Similarly, why is the charge transfer time in **Ph(Tc-Py-H⁺)** slower in benzonitrile than in toluene, despite the faster decay of the emissive CT state, as expected? For example, is this because of solvent gating or an energy (Marcus) effect?

Author reply: We thank the reviewer for this insightful comment and agree that it is important to establish the photophysics of the protonated tetracene monomers and dimers. As implied by the reviewer, studies on charge transfer dynamics have been examined in terms of Marcus theory, which successfully explained photoinduced CT dynamics in the Marcus normal and/or inverted regions. Notably, the Marcus formalism is applied to weakly coupled electron donor-acceptor systems. In **Ph(Tc-Py-H⁺)**, the ethynyl bridge between tetracene and pyridinium is the basis for strong coupling (electronic communication). Hence, the electron donor and acceptor in **Ph(Tc-Py-H⁺)** are not in a weak but in a strong coupling regime. Therefore, adiabatic charge transfer, in which the nuclear motion is coupled to electron motion, and, in which the system remains on one surface, should be considered in our case (10.1021/jacs.9b12016; 10.1002/cphc.201900703;10.1021/acs.jpcc.9b07077). Moreover, charge transfer processes in **Ph(Tc-Py-H⁺)** occur rapidly in a few picoseconds regardless of the solvent (ca. 10 ps in benzonitrile and ca. 8 ps toluene). This fact confirms that any charge transfer in **Ph(Tc-Py-H⁺)** is adiabatically following the solvent relaxation dynamics. Here, the transition from a locally-excited state (LE) to CT is influenced by the solvent polarity, solvent friction, and coordination of solvent and nuclear reactions, etc. Differences for the LE-to-CT transition between benzonitrile (ca. 10 ps) and toluene (ca. 8 ps) are likely due to solvent viscosity. We rationalize this observation by the greater viscosity of benzonitrile, which slows structural relaxation and solvation. Moreover, the CT state(s) in toluene and benzonitrile are both bright states. This indicates that a dark charge-separated state is not formed in **Ph(Tc-Py-H⁺)**. Subsequently, for **Ph(Tc-Py-H⁺)**, charge recombination takes place. Considering the red-shifted emission observed in polar benzonitrile, the CT state in benzonitrile is located on a lower energy adiabatic potential energy surface. From the experimental results, the lifetime of charge recombination for **Ph(Tc-Py-H⁺)** is ca. 1.2 ns in toluene and ca. 445 ps in benzonitrile. It is notable that the charge recombination is not ultrafast, and it exhibits a sizeable polarity-dependence. Thus, charge recombination is diabatic. The smaller energy gap between the CT state and the ground state thus increases the driving force for charge recombination in polar solvent, leading to a faster charge recombination in benzonitrile. We have revised this discussion in the new version of the manuscript to clarify the excited state dynamics of **Ph(Tc-Py-H⁺)** and revised the schematics regarding the potential energy surface of **Ph(Tc-Py-H⁺)** as shown in Figure R1.

Please see the revised version of the manuscript: Figure 6b highlighted in yellow.

Figure R1 (Figure 6b). Proposed schematic potential energy surface of **Ph(Tc-Py-H⁺)** and **mPh(Tc-Py-H⁺)₂** in which (S₁) is in red, intra-CT state is in blue (dark blue represents intra-CT state in benzonitrile and sky blue represents intra-CT state in toluene), and (S₀) is in black.

Please see the revised version of the manuscript: lines 300–311 highlighted in yellow.

It is observed that the intra-CT state in **Ph(Tc-Py-H⁺)** forms rapidly within a few picoseconds, regardless of the solvent. This rapid formation is attributed to the strong coupling between the pyridinium and tetracene moieties through the acetylene spacer, which enables an adiabatic charge transfer that follows solvent relaxation dynamics.^{82–84} In the more viscous solvent benzonitrile, slower structural relaxation and solvation dynamics lead to a slightly delayed formation of the intra-CT state. Subsequently, in benzonitrile, the intra-CT state is shorter-lived and decays back to the ground state within 445.7 ps. Charge recombination (CR) in **Ph(Tc-Py-H⁺)** is slow and is subject to a strong polarity dependence. The lifetime of intra-CT state in non-polar toluene (1.2 ns) is thus longer than in more polar benzonitrile (445.7 ps). Overall, CR occurs diabatically. In polar solvents like benzonitrile, the smaller energy gap between the intra-CT state and the ground state increases the driving force for charge recombination and expedites it.

Reviewer query: It would be useful if the authors can give a little more discussion about the yields of the various triplet states. The experimental estimations of the T₁+T₁ yields from oxygen sensitization are quite helpful as lower bounds, but can they be correlated with estimates of the yields from kinetic analysis? This might be difficult given the mixed nature of the S₁S₀/CT/TT state, but that intermediate appears to be formed in very high yield. The experimental estimates for the T₁+T₁ yields would then seem to imply significant spin evolution prior to decorrelation, potentially involving the 3TT state that diminishes the overall uncorrelated triplet yield. Without direct probing of these effects by spin-sensitive techniques such discussion will admittedly be speculative, but it may point to loss mechanisms that could be considered in future designs.

Author reply: We thank the reviewer for these suggestions and fully agree that it is difficult to determine the triplet pair quantum yields from only a kinetic analysis. A well-established

approach is to take advantage of the ground state bleaching intensification during the formation of the correlated triplet pair (10.1021/jacs.8b09510; 10.1021/jacs.2c13353). In the case of tetracene, however, this method is challenged by the overlap of the ground state bleaching (GSB), stimulated emission (SE), and excited state absorptions (ESA). Additionally, in our case, the mixed-state characterized by overlapping signatures of the singlet excited state, charge transfer state, and triplet excited state serves as an intermediate state throughout singlet fission, and the mixed state is not fully deconvoluted on the time scale of fs-TAS. Moreover, its rapid formation occurs beyond the resolution of ns-TAS. As a result, neither fs-TAS nor ns-TAS can be used for the determining the triplet pair quantum yield by kinetic analyses. As an alternative, we have used singlet oxygen quantum yields as an indirect measure of tetracene triplet excited states, since the (T_1) energy of tetracene (1.2 eV) is higher than that of O_2 (0.98 eV). But, we do acknowledge that the yields from this indirect method are not limited to the yields of pure free triplet excited because the triplet pair and mixed state are also likely to be quenched by O_2 (10.1038/s41467-019-13202-5; 10.1021/jacs.3c12245).

We agree with the reviewer that the discussion of dissociation in the singlet fission process is important for future design. For **mPh(Tc-Py)₂** in toluene, we are able to derive the dissociation yield (Φ_{Diss}) of the triplet pair. This analysis is supported by the invariance of the ground state bleaching and triplet excited state absorption extinction coefficients throughout the dissociation process. The dissociation yields have been determined by calculating the ratio of the ΔOD values that correspond to the evolution-associated spectra of (T_1T_1) and ($T_1 + T_1$). Based on this fitting, Φ_{Diss} of **mPh(Tc-Py)₂** in toluene is low. The value is around 8%. Such a low Φ_{Diss} is consistent with observations made for other tetracene and pentacene dimers with meta-phenylene spacers. For instance, the tetracene dimer (**mPhTc₂**) shows a Φ_{Diss} of 5%, the pentacene dimer (**m-2**) exhibits no detectable free triplet excited state, and the pentacene dimer (**TIBS-Pnc₂**) has a Φ_{Diss} of around 1% (10.1021/jacs.3c02417; 10.1073/pnas.142243611; 10.1002/anie.202315064).

For singlet fission in molecular dimers, efficient triplet dissociation, namely the formation of ($T_1 + T_1$), relies on the quintet of the correlated triplet pair $^5(T_1T_1)$ as the key intermediate. This process requires significant mixing of $^1(T_1T_1)$ and $^5(T_1T_1)$ (10.1021/cr1002613; 10.1016/j.chempr.2018.04.006; 10.1038/ncomms15171; 10.1073/pnas.1820932116; 10.1016/j.chempr.2019.05.012). Notably, such mixing is only feasible when the interchromophoric coupling is weak. In tetracene and pentacene dimers linked by meta-phenylene spacers, strong interchromophore coupling disrupts the degeneracy of $^1(T_1T_1)$ and $^5(T_1T_1)$ and inhibits effective mixing. As a result, (T_1T_1) dissociation is unlikely in these dimers, and triplet-triplet annihilation back to the ground state dominates. Another pathway is active in meta-tetracene dimer **mPh(Tc-Py)₂**: up-conversion to regenerate (S_1S_0). We agree with the reviewer that $^3(T_1T_1)$ might also trigger the free triplet (T_1S_0) state formation. However, $^1(T_1T_1)$ - $^3(T_1T_1)$ mixing is spin-forbidden (10.1016/j.chempr.2019.05.012).

Accordingly, we have added the discussion on spin-mixing in the revised version of the manuscript to improve the clarity and to avoid any further confusions.

Please see the revised version of the manuscript: lines 345–356 highlighted in yellow.

It is important to note that Φ_{Δ} does not represent the yield of a pure free triplet excited state, since both (T_1T_1) and mixed states are likely to be quenched by O_2 .^{88–90} The dissociation yield (Φ_{Diss}) of the triplet pair state was determined by calculating the ratio of ΔOD values corresponding to the evolution-associated spectra of (T_1T_1) and ($T_1 + T_1$). Accordingly, Φ_{Diss} of **mPh(Tc-Py)₂** in toluene is low, ca. 8%, which is similar to previous reports for meta-phenylene-linked tetracene and pentacene dimers.^{43,54} The strong electronic coupling between the two tetracenes moieties through the meta-phenylene spacer enhances exchange interactions between $^1(T_1T_1)$ and $^5(T_1T_1)$, consequently suppresses spin-mixing of $^1(T_1T_1)$ - $^5(T_1T_1)$, which ultimately renders dissociation of (T_1T_1) highly unlikely.^{19,40,41,91,92} Moreover, the evolution of triplet excited state for **mPh(Tc-Py)₂** does not on the photoexcitation wavelength (Figure S22), suggesting a rigid and structurally homogeneous conformation in solution, consistent with the observation in steady state absorption analysis.

Reviewer query: Are the TRES component spectra observed for mPh(Tc-Py)₂ identical, or are there subtle shifts that might imply alterations in the state energies during the mixing process? Normalization of the component spectra in Figure S22b will address this.

Author reply: We thank the reviewer for this suggestion. In the revised version of the supporting information, we have added the normalized component spectra as an inset in Figure S27 (Figure R2). Notably, the deconvoluted component spectra observed in **mPh(Tc-Py)₂** are identical. Given that no spectral changes are discernible throughout TRES, all three emission components are attributed to a single emissive state. Importantly, the TRES lifetimes match well those of the (S_1S_0)_{rel}, (M), and (T_1T_1) states, respectively, in the TAS measurements. Furthermore, due to the slight exothermicity of triplet-triplet annihilation upconversion in tetracenes, it is plausible for (T_1T_1) to upconvert to (S_1S_0)_{rel} and then, subsequently, decay (10.1039/C8SC03725F; 10.1021/acs.jpcclett.9b03115; 10.1021/jacs.0c06386). Therefore, the prompt fluorescence comes directly from (S_1S_0)_{rel}, while the two delayed fluorescence events arise from the recombination of the (M) state and triplet-triplet annihilation of the (T_1T_1) state, respectively, to the (S_1S_0)_{rel}.

Please see the revised version of the supporting information: Figure S27 highlighted in yellow.

Figure R2 (Figure S27 a–c). TRES raw data and corresponding global analysis for **mPh(Tc-Py)₂**, following photoexcitation at 510 nm in argon-toluene at room temperature. (a). Heat map of the recorded TRES at various delays between 0 to 200 ns after photoexcitation at 510 nm. (b) Evolution-associated spectra (EAS); The normalized EAS in toluene are exhibited in the inset. (c) Relative populations of the respective species with colors correlating with the evolution-associated spectra (EAS)

Reviewer query: The IMPLICATION gate is claimed to be an "all-optical" logic gate (line 369). However, this seems odd because only the readout in this case is optical; the actions themselves are chemical. At the very least this should be clarified, but that particular claim seems like a stretch and should be downplayed.

Author reply: We appreciate this point and we apologize for the mistaken use of the term 'all-optical.' In the revised version of the manuscript, we have removed the description of 'all-optical' and emphasized the role of chemical input. Moreover, we agree with the comment that our IMPLICATION gate is simple, and we have deemphasized it. Accordingly, in the revised version of the manuscript, we have added previous investigations on controllable singlet fission and singlet fission switches, which adds context to the novelty of our project. We have utilized chemical input (acid/base stimuli) to achieve a simple IMPLICATION logic gate operation within singlet fission. However, we believe that more complex logic gate operations in singlet fission should be realized in future investigations. We hope that our work will serve as a starting point for more sophisticated logic operations based on singlet fission. We have included a discussion of these perspectives in the conclusion and outlook sections of the revised version of the manuscript.

Please see the revised version of the manuscript: lines 451–453 highlighted in yellow

The switchable, reversible SF behavior of **mPh(Tc-Py)₂** toward TFA and TEA encouraged us to consider its application as an IMPLICATION gate using chemical input (TFA/TEA) (Figure 6c).

Please see the revised version of the manuscript: lines 52–63 highlighted in yellow.

Singlet fission (SF) can, in principle, convert a singlet exciton into two triplet excitons. Thus, the maximum triplet quantum yield for SF is 200%.^{19–23} SF is a well-known multiple-exciton generation process that is of interest not only for efficient solar energy harvesting^{24–26} but also for photocatalysis,²⁷ photodynamic oncotherapy,²⁸ large optical nonlinearity.^{29–32} Observing a high-spin quintet state in SF, which is undoubtedly a rarity in organic materials, recently invokes the potential for SF to be applied in quantum information science.^{33–35} In short, designing a molecular logic gate based on an SF switch would not only enable on/off control over multi-excitonic processes but also open up new avenues for exploring triplet excited-state switching. Therefore, it would be beneficial to realize a SF switch that responds to environmental stimuli and that transforms SF beyond basic research to, for example,

molecular computers, molecular mechanisms, and biology as achieved by traditional molecular switches.^{1,3,4,36}

Please see the revised version of the manuscript: lines 84–106 highlighted in yellow.

Previous research efforts have been dedicated to controlling SF, and, in turn, these studies form the foundation for design of a SF switch. In the solid state, SF is mostly investigated in monomers with variable molecular packing through molecular engineering^{57–62} or conditioned fabrication.^{63,64} In the context of molecular engineering, tunable SF is investigated in different stacking geometries. These are adjusted by functionalized SF chromophores, such as alkyl-substituted terylenes⁵⁹ and diketopyrrolopyrroles.^{57,60} For conditioned fabrication, SF processes are compared in amorphous films, polycrystalline films, and single crystals. Moreover, changes in SF in films are reported upon varying the aggregation through varying the solvent or different ratios of the doped polymer matrix.⁶² Controlling the packing modes and achieving homogeneity in the solid state is, per se, challenging and limits utility as a platform for the design of SF switches. In contrast, molecular dimers offer advantages such as studies in solution with precise control and tuning of the molecular structure, as well as the distance, geometric relationship, and electronic coupling between two chromophores. For example, Alvertis and coauthors demonstrated that the transition between an incoherent and a coherent mechanism for SF in an orthogonal tetracene dimer is modulable by varying the solvent environment.⁶⁵ Wasielewski and coworkers work with terylenediimide dimers noted that the CT state in polar solvents not only acted as a trap state but also suppressed SF.⁴⁹ Guldi and coworkers developed the concept of SF switching by employing a diamantane spacer placed between two pentacenes. They showed that SF could be toggled 'on' or 'off' by altering the substitution pattern.³⁷ Thus, they realized a SF switch through two distinct dimeric isomers. Altogether, these studies demonstrate that SF is sensitive to both molecular structure and the environment. To the best of our knowledge, however, a reversible SF switch in a molecular system, which can be further applied to molecular logic gates, has never been reported.

Please see the revised version of the manuscript: lines 487–495 highlighted in yellow

In this project, we utilized acid and base as stimuli to realize the triplet pair/SF switch and molecular logic gate. Beyond this, other inputs, such as optical signals, electronic signals, and magnetic signals, hold significant potential for the development of more diverse SF switches. Notably, future investigations could address not only the 'on' and 'off' of SF, but also the controllable multiplicities of the triplet pair. Furthermore, the integration of multiple input signals could enable the design of more complex molecular logic gates based on SF. This work not only validates the viability of switchable multiple-exciton generation through a modulating CT state, but also opens the way for broader developing stimulus-responsive SF materials.

Reviewer query: Error bars/uncertainties should be presented for all values in the main text, and in the tables in the Supplementary Information.

Author reply: We appreciate this point, and we have incorporated the uncertainties in the Methods section of the revised version of the manuscript and in the revised version of the supporting information accordingly. For parameters that have been determined via transient absorption measurements, we consider an error in the range of $\pm 10\%$, as this is the typical error in our photophysical measurements. For extinction coefficients, we give of $\pm 8\%$, which was verified by multiple measurements of the extinction coefficients. The fluorescence quantum yields have been determined via an absolute method using an integrating sphere setup and the uncertainties of 5–10% have been used as reported (10.1038/nprot.2013.087). For the current measure we have used a conservative error of $\pm 10\%$.

Please see the revised version of the manuscript: lines 517–521 highlighted in yellow

For molar extinction coefficients, repeated measurements verified an error in the range of $\pm 8\%$. The fluorescence quantum yields were determined via an absolute method using an integrating sphere setup, and the uncertainties of 5–10% can be achieved as reported.⁹³ Here, the error of $\pm 10\%$ for absolute fluorescence quantum yield is considered.

Please see the revised version of the manuscript: lines 543–545 highlighted in yellow

For parameters determined via transient absorption measurements, an error in the range of $\pm 10\%$ is considered, as this is the typical error in photophysical measurements.

Reviewer query: A kinetic scheme for the models used and the energies of the states in the dimer would be useful, even in the Supplementary Information.

Author reply: We thank the reviewer for this suggestion and agree that implementing a kinetic scheme is helpful. We have added it for the models in the revised version of the supporting information.

Please see the revised version of the supporting information: Figure S25 highlighted in yellow

Figure R3 (Figure S25). Qualitative energy diagram showing the sequential deactivation cascades for mPh(Tc-Py)₂ in (a) toluene and (b) benzonitrile after excitation at 510 nm. For mPh(Tc-Py)₂ in toluene, initial photoexcitation into (S_1S_0) is followed by solvent relaxation to produce (S_1S_0)_{rel}, which is subsequently converted into a mixed state (M) which is a combination of the (S_1S_0)_{rel}, CT, and (T_1T_1) states. Then, the correlated triplet pair (T_1T_1) is formed via the mixed state. Next, the spin decoherence of (T_1T_1) yields free triplet excited states (T_1+T_1). For mPh(Tc-Py)₂ in benzonitrile, the initial excitation into (S_1S_0) is followed by the formation of an initial mixed state (M_1). Subsequently, another mixed state (M_2), with the more intense signature of the CT state, is formed. Next, a free triplet excited state (T_1+T_1) is followed without intermediation of a pure (T_1T_1) state.

Minor issues:

Reviewer query: Line 210: "The kinetic model of the ns-TAS data..." The sentence goes on to discuss time constants on the order of ~ps, indicating that this is actually the "fs-TAS" data.

Author reply: We apologize for the error. It has been corrected accordingly in the revised version of the manuscript.

Please see the revised version of the manuscript: line 249-251 highlighted in yellow.

The kinetic model for the fs-TAS data involves (S_1), which undergoes solvent reorganization to afford (S_1)_{rel} within 3.1 ps and then completely reinstates the ground state (S_0) within 8.2 ns.

Reviewer query: Line 361 is confusingly worded, as it seems to imply that the deprotonation is being monitored on the ultrafast timescale. Please revisit.

Author reply: We have rewritten this sentence in the revised version of the manuscript.

Please see the revised version of the manuscript: line 441-443 highlighted in yellow.

In fs-TAS analysis of **mPh(Tc-Py)₂** after the sequential addition of TFA and TEA, the characteristics of (T_1T_1) at 470 nm are observed within hundreds picoseconds along with the characteristics of (S_1S_0)_{rel} in the range of 600–750 nm.

Reviewer query: There are several typos in the SI: Figures S14 and S16 both say they present TAS data for the NIR region despite showing data for the visible spectrum.

Author reply: We thank the reviewer for pointing out these typos. We have corrected it in the revised version of the supporting information.

Please see the revised version of the supporting information: Figure S15 and S17 (Figure S14 and S16 in the original version of supporting information) highlighted in yellow.

Figure S15. Nanosecond transient absorption spectroscopy **in visible region of Ph(Tc-Py)** in argon-saturated toluene at room temperature. (a) Heat map of the recorded differential absorption spectra at various time delays between 0 to 330 μ s after photoexcitation at 510 nm. (b) Evolution-associated spectra (EAS) and insert that depicts time absorption profiles as well as corresponding fits of selected wavelengths (see Figure legend for details). (c) The relative population of the respective species with colors correlating with the evolution-associated spectra (EAS).

Figure S1. Nanosecond transient absorption spectroscopy **in visible region of Ph(Tc-Py)** in argon-saturated benzonitrile at room temperature. (a) Heat map of the recorded differential absorption spectra at various time delays between 0 to 330 μ s after photoexcitation at 510 nm. (b) Evolution-associated spectra (EAS) and insert that depicts time absorption profiles as well as corresponding fits of selected wavelengths (see the figure legend for details). (c) The relative population of the respective species with colors correlating with the evolution-associated spectra (EAS).

Reviewer #2 (Remarks to the Author):

Reviewer query: Bo et al. report on the singlet fission behavior of a tetracene dimer and the response of the photophysics to protonation/deprotonation of pendant amine groups. The authors show that protonation results in a significant enhancement in charge-transfer interactions, leading to a rapid CT-based decay pathway that outcompetes singlet fission. After deprotonation, singlet fission behavior (previously demonstrated in a related dimer) is restored. The basic spectroscopic behavior is clearly demonstrated, and the results are largely well explained. The study certainly merits publication, though I am not convinced that the novelty/significance rises to the level of Nature Communications: the basic concept is that chemical tuning that distorts the electronic structure will disrupt the photophysics (in this case singlet fission). It is very clearly demonstrated, but not particularly surprising. I have no major criticisms of the technical aspects of the work, however, which seem to be at a good level. Below I list some minor points that should be addressed prior to publication, for clarification or context.

Author reply: Thank you for all the careful comments and suggestions, and we seemed to have failed in our effort to emphasize the significance of our studies in the original version of the manuscript. In the revised version of the manuscript, we have expanded the discussion of relevant literature in the introduction section and have included a discussion of outlook in the conclusion section to emphasize the importance and future implications of our work, based on the following points.

Singlet fission is a photophysical down-conversion process, in which a singlet excited state is converted into two triplet excited states. As an efficient multiple exciton generation process in organic semiconductors, singlet fission holds significant potential, particularly in solar energy conversion. The initial purpose of studying singlet fission was to overcome the detailed balance limit in single-junction solar cells. Singlet fission has proven to have a broad potential in areas beyond photovoltaics, including photodynamic oncology (10.1021/acs.nanolett.4c01862), large optical nonlinearity (10.1063/5.0013985; 10.1039/c5mh00120j), quantum information, and molecular computing (10.1038/s41598-020-75459-x; 10.1063/5.0069344). Given this versatility, developing a singlet fission switch responsive to environmental stimuli would be transformative. Such a switch has the potential to extend singlet fission applications beyond basic research to practical implementations in molecular computers, mechanisms, and biological systems, akin to traditional molecular switches.

While some progress has been made toward the concept of a 'singlet fission switch', these efforts have largely been limited in scope. Alvertis and coauthors demonstrated switching between incoherent and coherent mechanisms by using different solvents (10.1021/jacs.9b05561). Wasielewski investigated tunable singlet fission in terrylenediimide dimers and concluded that the charge transfer state in polar solvents acts as a trap state, inhibiting the singlet fission (10.1038/nchem.2589). Guldi and Tykwinski developed the concept of a singlet fission switch by employing a singular spacer diamantane between two pentacene chromophores (10.1021/jacs.4c01507). They showed that singlet fission was toggled 'on' or 'off' by altering the substitution pattern in the two distinct dimeric isomers.

Although these studies highlight the sensitivity of singlet fission to environmental factors and molecular design, they lack reversibility in a sole molecular system.

To the best of our knowledge, our work is the first to achieve a reversible singlet fission switch in response to environmental stimuli. As noted by the reviewer, our fundamental concept is that chemical tuning by distorting the electronic structure changes the photophysics. Specifically, our molecular singlet fission switch reversibly transitions between two distinct charge transfer states. Using acid/base stimuli, a widely employed method in traditional molecular switches, we designed and realized this novel singlet fission switch system. Our study is just a starting point; more studies on molecular switches and complex molecular logic gates based on singlet fission are warranted. We feel that further inspiration will come from existing molecular switches, such as photochromic switches triggered by light irradiation, mechanically interlocked switches, and biological switches, to name just a few.

Overall, we believe that this study warrants recognition and dissemination within the scientific community, making our manuscript suitable for publication in the high-impact journal Nature Communications.

We have attempted to clarify the impact of our studies in the revisions.

Please see the revised version of the manuscript: lines 52–63 highlighted in yellow.

Singlet fission (SF) can, in principle, convert a singlet exciton into two triplet excitons. Thus, the maximum triplet quantum yield for SF is 200%.^{19–23} SF is a well-known multiple-exciton generation process that is of interest not only for efficient solar energy harvesting^{24–26} but also for photocatalysis,²⁷ photodynamic oncotherapy,²⁸ large optical nonlinearity.^{29–32} Observing a high-spin quintet state in SF, which is undoubtedly a rarity in organic materials, recently invokes the potential for SF to be applied in quantum information science.^{33–35} In short, designing a molecular logic gate based on an SF switch would not only enable on/off control over multi-excitonic processes but also open up new avenues for exploring triplet excited-state switching. Therefore, it would be beneficial to realize a SF switch that responds to environmental stimuli and that transforms SF beyond basic research to, for example, molecular computers, molecular mechanisms, and biology as achieved by traditional molecular switches.^{1,3,4,36}

Please see the revised version of the manuscript: lines 84–106 highlighted in yellow.

Previous research efforts have been dedicated to controlling SF, and, in turn, these studies form the foundation for design of a SF switch. In the solid state, SF is mostly investigated in monomers with variable molecular packing through molecular engineering^{57–62} or conditioned fabrication.^{63,64} In the context of molecular engineering, tunable SF is investigated in different stacking geometries. These are adjusted by functionalized SF chromophores, such as alkyl-substituted terylenes⁵⁹ and diketopyrrolopyrroles.^{57,60} For conditioned fabrication, SF processes are compared in amorphous films, polycrystalline films, and single crystals. Moreover, changes in SF in films are reported upon varying the aggregation through varying the solvent or different ratios of the doped polymer matrix.⁶² Controlling the packing modes and achieving homogeneity in the solid state is, per se,

challenging and limits utility as a platform for the design of SF switches. In contrast, molecular dimers offer advantages such as studies in solution with precise control and tuning of the molecular structure, as well as the distance, geometric relationship, and electronic coupling between two chromophores. For example, Alvertis and coauthors demonstrated that the transition between an incoherent and a coherent mechanism for SF in an orthogonal tetracene dimer is modulable by varying the solvent environment.⁶⁵ Wasielewski and coworkers work with terrylenediimide dimers noted that the CT state in polar solvents not only acted as a trap state but also suppressed SF.⁴⁹ Guldi and coworkers developed the concept of SF switching by employing a diamantane spacer placed between two pentacenes. They showed that SF could be toggled 'on' or 'off' by altering the substitution pattern.³⁷ Thus, they realized a SF switch through two distinct dimeric isomers. Altogether, these studies demonstrate that SF is sensitive to both molecular structure and the environment. To the best of our knowledge, however, a reversible SF switch in a molecular system, which can be further applied to molecular logic gates, has never been reported.

Please see the revised version of the manuscript: lines 487–495 highlighted in yellow

In this project, we utilized acid and base as stimuli to realize the triplet pair/SF switch and molecular logic gate. Beyond this, other inputs, such as optical signals, electronic signals, and magnetic signals, hold significant potential for the development of more diverse SF switches. Notably, future investigations could address not only the 'on' and 'off' of SF, but also the controllable multiplicities of the triplet pair. Furthermore, the integration of multiple input signals could enable the design of more complex molecular logic gates based on SF. This work not only validates the viability of switchable multiple-exciton generation through a modulating CT state, but also opens the way for broader developing stimulus-responsive SF materials.

Reviewer query: 1. The authors should provide normalized EAS for comparison in the SI. For instance, in line 271-272 they note that the fourth species 'does not display the signatures of (S1S0)'. From the text or figure it is not immediately clear which signatures these are (what wavelengths/peaks are key to this analysis?), while the intensity differences between different EAS make it extremely difficult to evaluate what, if anything distinguishes the species.

Author reply: We thank the reviewer for these suggestions and agree that providing normalized EAS would help the readers to follow the description in the manuscript. Accordingly, we have added the normalized EAS in the revised version of the supporting information. In that way, it should be easier for the reader to follow the discussion about the signatures of EAS.

Please see the revised version of the supporting information: Figure S21 and S24 highlighted in yellow

Figure S21. Normalized evolution-associated spectra (EAS) of the respective species for nanosecond transient absorption spectroscopy of **mPh(Tc-Py)₂** in argon-saturated toluene at room temperature.

Figure S24. Normalized evolution-associated spectra (EAS) of the respective species for nanosecond transient absorption spectroscopy of **mPh(Tc-Py)₂** in argon-saturated benzonitrile at room temperature.

Reviewer query: 2. In the global analysis of the excited-state dynamics, the authors require a large number of components (4 or 5 depending on solvent) to accurately represent the data. This appears to be because of the choice of a sequential model. Some of the authors, as well as others in the field (e.g. Ringstrom et al., DOI: 10.1039/D1SC06285A and Kim et al., DOI: 10.1016/j.xcrp.2024.102045), have shown that such acetylene-bridged acene dimers inevitably exhibit significant rotational disorder. Is a possible alternative explanation of the reported dynamics of the variously singlet-CT-TT mixed states that they are capturing the evolution of a disordered ensemble, effectively a branched/parallel decay pathway?

Author reply: We appreciate these suggestions on the potential of branched/parallel modes of the TA data. It is true that some researchers embrace a branched/parallel model of the global target analysis on singlet fission investigation of conformational heterogeneous systems. For the conformational heterogeneous singlet fission systems, the steady-state absorption spectra of the dimers exhibit broadening effects or additional absorption bands, contrary to the well-defined spectra of respective monomers (Figure R4b and R4c) (10.1039/D1SC06285A; 10.1016/j.xcrp.2024.102045). However, compared to the absorption spectra of the monomer **Ph(Tc-Py)**, the dimer **mPh(Tc-Py)₂** shows slight red-shifts due to the π -extension, but not additional absorptions (Figure R4a). Most importantly, the absorption spectrum of **mPh(Tc-Py)₂** lacks broadening typical of conformational heterogeneity, and the

bandwidth of the dimer is comparable to that of the monomer. Finally, wavelength-dependent photoexcitation in the fs- and ns-TAS has been carried out with **mPh(Tc-Py)₂** in toluene. The triplet excited state signature of **mPh(Tc-Py)₂** does not demonstrate dependence on the photoexcitation wavelength, neither with respect to formation nor decay (Figure R5a and R5b), which is different from the singlet fission systems that need branched or parallel decay pathways as shown in Figure R5 c–f. Therefore, we have excluded conformational heterogeneity as a variable in the dynamics of **mPh(Tc-Py)₂** and focused on the sequential model analysis for singlet fission. To clarify this, we have added this discussion in the revised version of the manuscript.

Figure R4. (a) The normalized absorption spectra of **Ph(Tc-Py)** and **mPh(Tc-Py)₂**. (b–c) A comparison between monomer and dimer in other publications: (b). DOI: 10.1016/j.xcrp.2024.102045; (c) DOI: 10.1039/D1SC06285A

Figure R5. (a) Femtosecond transient absorption single wavelength kinetics of the triplet excited state signature at 473 nm of $mPh(Tc-Py)_2$ in Ar-saturated toluene at different photoexcitation wavelengths 390, 510, and 570 nm; (b) nanosecond transient absorption (ns-TAS) single wavelength kinetics of the triplet excited state signature at 473 nm of $mPh(Tc-Py)_2$ in Ar-saturated toluene at photoexcitation wavelengths of 390, 510, and 570 nm; (c–f) single wavelength kinetics of the triplet excited state signature for conformational heterogeneous singlet fission systems at different photoexcitation in other publications: (c) DOI: 10.1016/j.xcrp.2024.102045; (d–f) DOI: 10.1039/D1SC06285A.

Please see the revised version of the manuscript: lines 153–155 highlighted in yellow

Notably, the absorption spectrum of $mPh(Tc-Py)_2$ is well-defined without any discernable broadening compared to that of $Ph(Tc-Py)$, indicating the rigid and homogeneous conformation of $mPh(Tc-Py)_2$ in solution.

Please see the revised version of the manuscript: lines 353–356 and the revised supporting information: Figure S22 highlighted in yellow

Moreover, the evolution of triplet excited state for $mPh(Tc-Py)_2$ does not depend on the photoexcitation wavelength (Figure S22), suggesting a rigid and structurally homogeneous conformation in solution, consistent with the observation in steady state absorption analysis.

Figure S2. Triplet evolution dynamics of **mPh(Tc-Py)₂** in argon-saturated toluene at room temperature at different photoexcitation wavelengths of 390, 510, and 570 nm. (a) single-wavelength kinetics at 473 nm in femtosecond transient absorption spectroscopy. (b) single-wavelength kinetics at 473 nm in nanosecond transient absorption spectroscopy.

Reviewer query: 3. The authors describe the delayed emission once the system has evolved into the 1(T1T1) state as an upconversion process. But in Figure 6 the 1(T1T1) state is drawn distinctly higher in energy than S1S0, which would suggest a downconversion/relaxation. The description in the text and the schematic PES should be brought into agreement.

Author reply: We thank the reviewer for the comment and apologize for the misleading description. As reported, the energy of the singlet excited state (S_1) in tetracene is about twice that of the triplet excited state (T_1), leading to an isoergonic or slightly endergonic singlet fission in tetracenes. Accordingly, we have corrected Figure 6a in the revised version of the manuscript. To describe the back process from the mixed state and triplet pair state to the singlet excited state, we have corrected the description in the revised version of the manuscript.

Please see the revised version of the manuscript: lines 364–366 highlighted in yellow

In other words, the prompt fluorescence stems directly from $(S_1S_0)_{rel}$, while the two delayed fluorescence events arise from recombination involving (M) and (T_1T_1) , respectively.

Please see the revised version of the manuscript: Figure 6a highlighted in yellow

Figure R6 (Figure 6a). Proposed schematic potential energy surface of **mPh(Tc-Py)₂**, in which (S₁S₀) is in red, inter-CT state is in blue (dark blue represents inter-CT state in benzonitrile and sky blue represents inter-CT state in toluene), and ¹(T₁T₁) is in green.

Reviewer query: 4. Have the authors investigated a pump wavelength dependence? A previous quite relevant study on intramolecular singlet fission in a tetracene dimer seems to have been missed: Alvertis et al., DOI: 10.1021/jacs.9b05561. That work also invoked the idea of a ‘switch’, where the role and energy of charge-transfer states was a critical factor. In particular, the regime where a CT state served as a ‘trap’, as reported here in the TFA-treated dimer, could be apparently overcome by excitation at higher energy. It would be interesting to know if the logic behavior schematically portrayed in Figure 6 changes with excitation conditions.

Author reply: We thank the reviewer for this intriguing suggestion. First, we apologize for overlooking the work by Alvertis et al., which provides an excellent reference on how the CT state influences singlet fission. In the revised version of the manuscript, we have included discussion regarding this work, along with other relevant studies.

We also agree with the reviewer that it would be interesting to establish changes under different photoexcitation conditions to realize even more complex logic behavior. In the tetracene dimer from Alvertis et al., competition between populating an emissive locally excited state (LE) and an emissive charge state is proposed (10.1021/jacs.9b05561). As shown in the excitation and emission map (Figure R7b), different LE-to-CT ratios are formed under different photoexcitation conditions. The singlet fission mechanism is also affected. Only the emissive CT state is observed for **mPh(Tc-Py-H⁺)₂** in the excitation and emission map (Figure R7a). From this observation, we infer that the lower-energy CT state, which is a trap state for singlet fission, is populated even under high-energy photoexcitation.

We have performed pump wavelength-dependent fs-TAS measurement with **mPh(Tc-Py-H⁺)₂** in toluene, and we have not observed any changes in the kinetics with different pump wavelengths (Figure R8). We have added these details to the revised version of the manuscript. Nevertheless, the reviewer raises a good point regarding the potential of optical inputs to drive more complex logic behavior in singlet fission switches.

Figure R7. (a) Excitation and emission map for $m\text{Ph}(\text{Tc-Py-H}^+)_2$ in toluene. (b) Excitation and emission map for DT-Mes in DOI: 10.1021/jacs.9b05561.

Please see the revised version of the manuscript: lines 222–227 highlighted in yellow

Moreover, as seen in excitation-emission maps that were recorded for $m\text{Ph}(\text{Tc-Py-H}^+)_2$ in toluene (Figure S13), intra-CT emission dominates. This occurs regardless of the photoexcitation wavelength and locally-excited emission is not observed even under high-energy photoexcitation. This emission behavior indicates that the formation of a low-lying intra-CT state is highly competitive.

Please see the revised version of the supporting information: Figure S13 highlighted in yellow

Figure S13. Excitation and emission map for $m\text{Ph}(\text{Tc-Py-H}^+)_2$ in toluene.

Please see the revised version of the manuscript: lines 433–435 highlighted in yellow

After protonation, $m\text{Ph}(\text{Tc-Py-H}^+)_2$ exhibits the same excited-state dynamics as $\text{Ph}(\text{Tc-Py-H}^+)$, and, in particular, the formation of triplet excited states is absent, regardless of the solvent or photoexcitation wavelength (Figures S28–30).

Please see the revised version of the supporting information: Figure S28 highlighted in yellow

Figure R8 (Figure S28). Femtosecond transient absorption spectroscopy of **mPh(Tc-Py-H⁺)₂** in argon-saturated toluene at different photoexcitation wavelengths of 390, 510, 570, and 630 nm: single wavelength kinetics at (a) 461, (b) 636, and (c) 673 nm.

Reviewer query: 5. There is an extra Ph(Tc-Py-H+) on line 227.

Author reply: We thank the reviewer for pointing out this typo. We have deleted it in the revised version of the manuscript.

Reviewer #3 (Remarks to the Author):

The authors describe an acid/base stimuli-responsive conjugate for singlet fission. They show convincingly that charge-transfer states have a direct impact on the occurrence of singlet fission. While the non-protonated conjugate shows singlet fission, protonation leads to an energetically stabilized charge-transfer state that acts as a sink thereby avoiding singlet fission. The photophysical description of the involved pathways and transients is nicely supported by in-depth steady-state and time-resolved spectroscopic experiments.

Author reply: We thank the reviewer for the positive comments and recommendation of our work.

Reviewer query: The stimuli-responsiveness is summarized as a binary logical operation (IMP gate). This one I would not call “an all-optical” gate, because the input signals are not optical signals.

Author reply: We appreciate this comment (which was also pointed out by Reviewer #1) and apologize for the mistaken use of the term ‘all-optical.’ We agree that the input signals are not optical signals but chemical stimuli. We have deleted the description in the revised version of the manuscript.

Please see the revised version of the manuscript: lines 451–453 highlighted in yellow

The switchable, reversible SF behavior of **mPh(Tc-Py)₂** toward TFA and TEA encouraged us to consider its application as an IMPLICATION gate using chemical input (TFA/TEA) (Figure 6c).

Reviewer query: The work indeed opens new avenues for the design of switchable singlet-fission materials. Conceptually the use of acid and base is fine and the performance for a reduced number of switching cycles seems reasonable. However, in the long term the repeated acid-base neutralization will accumulate salt waste, which may influence the overall polarity of the medium (which ultimately would have impact of the singlet fission efficiency). Maybe this could be mentioned in the paper.

Author reply: We thank the reviewer for this excellent point and agree that the long-term accumulation of salt waste will affect the singlet fission. We have added this discussion to the revised version of the manuscript.

Please see the revised version of the manuscript: lines 195–197 highlighted in yellow

It should be noted that the accumulation of byproducts generated from acid and base additives is likely to hinder the complete resetting of chemical arithmetic systems, thereby disrupting the overall reversibility that is required for cyclic operations.^{69–71}

Please see the revised version of the manuscript: lines 446–450 highlighted in yellow

From fs-TAS, it is shown, for example, that contributions from $(S_1S_0)_{rel}$ are increased in the mixed state (species 3 in Figure S31), likely due to the increased polarity of solvent from the increasing concentration of salts from acid and base inputs. Thus, the complete resetting of the environment is hindered by salt accumulation, which affects the perfect reversibility of SF switches.

Reviewer query: The subtle interplay between charge-transfer states as “mediators or inhibitors” of the singlet-fission process is a nice idea and for providing an experimental demonstration of this idea the manuscript is a strong candidate for publication in Nat. Commun.

Author reply: We appreciate the kind recommendation of the reviewer.

Reviewer query: The translation of the experimental switching into an IMP logic is correct, but from my point of view it does not add too much to the paper. The herein described logic feature is comparably simple. Molecular logic (with chemical, optical or other inputs) has simulated much more complex logic operations in the past. Therefore, I would advice to base the introductory focus on the conceptual novelty of the switching process itself rather than on the resemblance with logic operations.

Author reply: We appreciate this insightful suggestion and agree that focusing on the singlet fission switch itself is essential. To address this in the revised version of the manuscript, we have added previous investigations on controllable singlet fission and singlet fission switches and compared these studies with our current work to highlight the overall novelty.

We thank the reviewer for pointing out more complex logic operations in the previous molecular switch investigation. Our study employs acid/base stimuli, inspired by prior molecular switch studies, to achieve a simple IMPLICATION logic gate operation in singlet fission. We fully agree that the field should expand beyond such stimuli and a simple logic gate. For singlet fission switches or logic gates, exploring alternative inputs such as electronic signals, magnetic signals, or a combination of multiple input signals, as already demonstrated is an exciting and likewise challenging direction for future research. We do hope that our work

serves as a starting point for exploring diverse switches and more complex logic operations based on singlet fission.

In the revised version of the manuscript, we have expanded the discussion of the relevant literature in the introduction section and have included some discussions regarding an outlook in the conclusion section to emphasize the importance of our work and future implications thereof.

Please see the revised version of the manuscript: lines 52–63 highlighted in yellow.

Singlet fission (SF) can, in principle, convert a singlet exciton into two triplet excitons. Thus, the maximum triplet quantum yield for SF is 200%.^{19–23} SF is a well-known multiple-exciton generation process that is of interest not only for efficient solar energy harvesting^{24–26} but also for photocatalysis,²⁷ photodynamic oncotherapy,²⁸ large optical nonlinearity.^{29–32} Observing a high-spin quintet state in SF, which is undoubtedly a rarity in organic materials, recently invokes the potential for SF to be applied in quantum information science.^{33–35} In short, designing a molecular logic gate based on an SF switch would not only enable on/off control over multi-excitonic processes but also open up new avenues for exploring triplet excited-state switching. Therefore, it would be beneficial to realize a SF switch that responds to environmental stimuli and that transforms SF beyond basic research to, for example, molecular computers, molecular mechanisms, and biology as achieved by traditional molecular switches.^{1,3,4,36}

Please see the revised version of the manuscript: lines 84–106 highlighted in yellow.

Previous research efforts have been dedicated to controlling SF, and, in turn, these studies form the foundation for design of a SF switch. In the solid state, SF is mostly investigated in monomers with variable molecular packing through molecular engineering^{57–62} or conditioned fabrication.^{63,64} In the context of molecular engineering, tunable SF is investigated in different stacking geometries. These are adjusted by functionalized SF chromophores, such as alkyl-substituted terylenes⁵⁹ and diketopyrrolopyrroles.^{57,60} For conditioned fabrication, SF processes are compared in amorphous films, polycrystalline films, and single crystals. Moreover, changes in SF in films are reported upon varying the aggregation through varying the solvent or different ratios of the doped polymer matrix.⁶² Controlling the packing modes and achieving homogeneity in the solid state is, per se, challenging and limits utility as a platform for the design of SF switches. In contrast, molecular dimers offer advantages such as studies in solution with precise control and tuning of the molecular structure, as well as the distance, geometric relationship, and electronic coupling between two chromophores. For example, Alvertis and coauthors demonstrated that the transition between an incoherent and a coherent mechanism for SF in an orthogonal tetracene dimer is modulable by varying the solvent environment.⁶⁵ Wasielewski and coworkers work with terylenediimide dimers noted that the CT state in polar solvents not only acted as a trap state but also suppressed SF.⁴⁹ Guldi and coworkers developed the concept of SF switching by employing a diamantane spacer placed between two pentacenes. They showed that SF could be toggled 'on' or 'off' by altering the substitution pattern.³⁷ Thus, they realized a SF switch through two distinct dimeric isomers. Altogether, these studies

demonstrate that SF is sensitive to both molecular structure and the environment. To the best of our knowledge, however, a reversible SF switch in a molecular system, which can be further applied to molecular logic gates, has never been reported.

Please see the revised version of the manuscript: lines 487–495 highlighted in yellow

In this project, we utilized acid and base as stimuli to realize the triplet pair/SF switch and molecular logic gate. Beyond this, other inputs, such as optical signals, electronic signals, and magnetic signals, hold significant potential for the development of more diverse SF switches. Notably, future investigations could address not only the 'on' and 'off' of SF, but also the controllable multiplicities of the triplet pair. Furthermore, the integration of multiple input signals could enable the design of more complex molecular logic gates based on SF. This work not only validates the viability of switchable multiple-exciton generation through a modulating CT state, but also opens the way for broader developing stimulus-responsive SF materials.

Response to Reviewers

Dear Reviewers:

Thank you for your valuable suggestions and kind recommendation of our work. We have addressed all comments by revising the Manuscript and provide a point-by-point response listed below. The changes in the revised version of the manuscript and the supporting information are highlighted in yellow, and the clarifications regarding the comments are provided in red.

Thank you again for your kind recommendation.

Dirk M. Guldi

Professor of Chemistry

Reviewer #1 (Remarks to the Author):

The authors have satisfactorily addressed my original concerns and I believe the revised scope and emphasis of the article make it suitable for publication in Nature Communications.

Author reply: Thank you for your valuable comments and suggestions, which are important in enhancing the quality of our paper to meet the publication standards of *Nature Communications*. Thank you again for your kind recommendation.

Reviewer #2 (Remarks to the Author):

Reviewer query: The authors have largely satisfied my concerns – the additional context and improved analysis are helpful.

Author reply: Thank you for all the careful comments and suggestions which have helped to improve the quality of our paper.

My only remaining concern is a minor one regarding the spectral decomposition. In the dimer, the authors invoke a state M in Figure 4 with a mixture of S₁, T₁T₁, and inter-CT character. While that is plausible in principle, in the accompanying text the authors only describe spectral features related to S₁ and T₁T₁. From close inspection of the figures, I concur that there is no identifiable CT character (it should manifest, for instance, as a difference in spectral shape between M and the final triplet state in the normalized spectra in the SI). Given there is no evidence for any involvement of CT character in the state M, I do not see how the authors can justify invoking it, and its mention should be removed. The authors' papers are widely read in the field, and it sets a dangerous precedent to make such assignments without direct support in the spectra.

Author reply: Thanks for the reviewer's suggestion. We apologize for the lack of clarity in our previous description. Firstly, the role of the mixed state in singlet fission has been well-established in various dimer systems and extensively reviewed in previous studies (DOI: 10.1021/jacs.3c02417; 10.1002/anie.202315064; 10.1021/jacs.2c13353; 10.1021/jacs.8b08627; 10.1021/jacs.8b04830; 10.1021/ar300286s). To further confirm the involvement of the inter-CT state in the singlet fission of mPh(Tc-Py)₂, we have examined and analyzed the dynamics of mPh(Tc-Py)₂ in the more polar solvent benzonitrile as presented in Figure 6 (previously Figure 5) of the manuscript and Supplementary Figures 23 and 24, along with their respective discussions. Regarding the evolution-associated spectra (EAS) of (M) in toluene, the contributions of an inter-CT state in apolar solvent toluene are weak. Additionally, the signatures of inter-CT state overlap with those of singlet and triplet excited-state, making it challenging to unambiguously identify its nature in toluene. However, the solvatochromic behavior of (M) and the presence of singlet and triplet excited-state markers in EAS of (M) confirm that the electronic nature of (M) in toluene by the mixing of (S₁S₀)_{rel}, CT state, and (T₁T₁), consistent with previous investigations.

Accordingly, we have added the discussion and cited references in the revised version of the manuscript to improve clarity and to avoid any further confusion.

Please see the revised version of the manuscript: lines 344–356 highlighted in yellow.

In line with previous studies,^{43,50,53,54,87,88} the third species is a mixed state (M), which is a superposition of (S₁S₀)_{rel}, CT state, and the triplet pair state (T₁T₁). Here, the CT state is the charge transfer between two tetracenes; we ascribe it to the interchromophore CT (inter-CT)

state. Notably, the polarity-dependent behavior of the mixed state suggest that the inter-CT state plays a role in the SF process (vide infra—the analysis of the excited state dynamics in benzonitrile). Furthermore, distinct signatures of inter-CT in the range of 800–1200 nm are clearly observed in the polar solvent benzonitrile. However, in toluene, the contributions of inter-CT in (M) are weak due to the higher energy of inter-CT in apolar solvents.^{49–51} Additionally, the spectral features of inter-CT overlap with the singlet and triplet excited-state absorptions, making its unambiguous identification in toluene challenging. Taking this into account, along with the solvatochromic nature of (M) and the presence of both singlet and triplet excited signatures in the EAS of (M), we conclude that the electronic nature of (M) by the mixing of states, consistent with previous investigation.^{43,54,88}

Reviewer #3 (Remarks to the Author):

The authors have presented a revised version of their manuscript, in which they have addressed my minor criticism in full. As far as I can see they have done also a very good job in revising their work regarding open points of their photophysical discussion (as suggested by my reviewer colleagues). From my point of view this insightful manuscript could be now accepted for publication.

Author reply: We appreciate your valuable comments and suggestions. We are grateful for your thoughtful recommendation. Thank you once again for your support.